



# Occurrence frequency of Kelvin Helmholtz instability assessed by global high-resolution radiosonde and ERA5 reanalysis

Jia Shao[1]; Jian Zhang[2]*; Wuke Wang[3]; Shaodong Zhang[4]; Tao Yu[2]; Wenjun Dong[5,6]

[1] College of Informatics, Huazhong Agricultural University, Wuhan 430070, China

[2] Hubei Subsurface Multi-scale Imaging Key Laboratory, School of Geophysics and Geomatics, China University of Geosciences, Wuhan 430074, China

[3] School of environmental studies, China University of Geosciences, Wuhan 430074, China

[4] School of Electronic Information, Wuhan University, Wuhan 430072, China

[5] Center for Space and Atmospheric Research (CSAR), Embry-Riddle Aeronautical University, Daytona Beach, FL, USA

[6] Global Atmospheric Technologies and Sciences (GATS), Boulder, CO, USA

Correspondence to:

Dr. Jian Zhang (Email: zhangjian@cug.edu.cn)





**Abstract.** Kelvin Helmholtz instability (KHI) is most likely to be the primary source for clear-air turbulence that is of importance in pollution transfer and diffusion and aircraft safety. It is exemplarily indicated by the critical value of Richardson ($Ri$) number, which is typically taken as 1/4. However, $Ri$ is fairly sensitive to the vertical resolution of the dataset: a higher resolution systematically leads to a finer structure. The study aims to evaluate the performance of ERA5 reanalysis (137 model levels) in determining KHI spatial-temporal variabilities, by comparing it against a near-global high-resolution (10-m) radiosonde dataset during years 2017 to 2022, and to further highlight the global climatology and dynamical environment of KHIs. Overall, the occurrence frequency of $Ri<1/4$ in the free atmosphere is inevitably underestimated by the ERA5 reanalysis over all climate zones, compared to radiosonde, due largely to the severe underestimation in wind shears. Otherwise, the occurrence frequency of KHI indicated by $Ri<1$ in ERA5 is climatologically consistent with that from radiosondes in the free troposphere, especially over the midlatitude and subtropics in the Northern/Southern Hemisphere. Therefore, we infer that the threshold value of $Ri$ should be approximated as 1, rather than 1/4, when using ERA5 for the KHI estimation. KHI occurrence frequencies revealed by both datasets exhibit significant seasonal cycles over polar, midlatitude, and subtropics regions, and they are consistently strong at heights of 10–15 km in the tropic region. In addition, the frequency at low-levels is positively correlated with the standard derivation of orography, and it is exceptionally strong over the Niño 3 region at heights of 6–13 km. Furthermore, the dynamical environment of KHI favors strong wind shears probably induced by the mean flows and the propagation of orographic or non-orographic gravity waves.

**Key words**: High-resolution radiosonde dataset; Kelvin Helmholtz instability; Threshold Richardson number; Gravity waves



## Introduction

Kelvin Helmholtz instability (KHI) is a common phenomenon in the atmospheric boundary layer and the free atmosphere (Muschinski and Wode, 1998), and its wavelengths and depths span a wide range of scales throughout the atmosphere, varying from few meters or less to 10s of km (Fritts et al., 2011). It contributes vertical mixing of heat, momentum, and constituents, and it acts to limit the maximum shears, just to name a few (Fritts et al., 2011). KHI along with gravity wave (GW) breaking are the most recognized instabilities in stably stratified flows (Fritts and Rastogi, 1985). In addition, GW breaking has been identified as important sources of instability (e.g., Fritts et al., 2020; Dong et al., 2020, 2021, 2022). KHI arises preferentially on strong shears due to medium-frequency and lower frequency GWs, tides, planetary waves (PWs), and mean flows (Baumgarten and Fritts, 2014). In addition, complex terrain may locally enhance wind shear, leading to KHI (Grasmick and Geerts, 2020). Large wind shear is common in regions where stability changes rapidly (e.g., near the top of the boundary layer) and the associated large gradient in jet stream (Grasmick and Geerts 2020), which may increase clear-air turbulence (Williams and Joshi, 2013). In turn, KHI can reduce wind shears and alter tracer gradients where turbulence and mixing are most intense (Fritts et al., 2022).

KHI influences depend on the spatial scales at which they lead to turbulence (Fritts et al., 2022). Turbulence is by far the most common cause of serious injuries to aircraft (Williams and Joshi, 2013). Convective instability, shear instability, KHI, and GW breaking are known to be the major source for turbulence (Sharman et al., 2012; Ko et al., 2019; 2022). Among others, KHI is one of the most common causes of turbulence throughout the atmosphere from Earth's surface to the lower thermosphere (Fritts et al., 2011; Sharman et al., 2012). KHI requires a sufficiently large Reynolds number and a Richardson ($Ri$) number sufficiently below 1/4 to enable KHI formation and subsequent secondary instability leading to turbulence (Fritts et al., 2022). $Ri$ is not a good guide to instability character in general, and $Ri>1/4$ does not assure flow

stability for superpositions of mean and GW motions. Despite these caveats, *Ri*<1/4 does provide a reasonable guide to expected local KHI structure in cases where clear KH billows arise (Fritts et al., 2014). Values of *Ri* close to zero favor strong instability, deep billows, and relatively intense turbulence, whereas values of *Ri* closer to 1/4 favor weak instability, shallow billows (Fritts et al., 2011). The threshold value of *Ri* can be potentially used an indicator of turbulence (for instance, Jaeger et al., 2007). Moreover, over half of turbulence exists below *Ri*<1 when the environment is beneficial for the development of turbulence (Zhang et al., 2022).

Turbulent mixing is of crucial importance to mass, energy, momentum transfer, the dispersion of pollutant, and stratosphere-troposphere exchange. However, it presents a challenge both in observation and numerical modeling (Sharman et al., 2012; Homeyer et al., 2014; Plougonven and Zhang, 2014). Due to the intermittent nature of turbulence it is generally not resolved in (global) numerical weather prediction models, even at nowadays common/states of the art horizontal resolutions of the order of tens of kilometers (Sandu et al., 2019).While in numerical models, turbulent dissipation rate, turbulent diffusivity and other parameters representing turbulent mixing efficiency are the most basic parameters, which need to be accurately parameterized to evaluate the impact of turbulent effect on matter and energy distribution (Gavrilov et al., 2005). For this reason, the indices of turbulence, such as large wind shear, small *Ri*, the negative squared Brunt-väisälä frequency, could be a great tool to characterize turbulence (Jaeger et al., 2007).

The *Ri* is estimated by the finite differences across thin layers and is quite sensitive to the vertical resolution of measurements (Haack et al., 2014). Thus, a proper estimation of *Ri* requires a high-resolution measurement of temperature and wind speed. The near-global distributed radiosonde site offers a unique opportunity to investigate the climatology of KHI occurrence frequency. The overview of KHI occurrence by using a near-global high-resolution (10-m) radiosonde data was presented in Zhang et al. (2022), and a close association between KHI occurrence frequency and turbulence fraction has been found. However, the global climatology characteristic of KHI remains most unclear, especially over oceans where the



radiosonde network has a poor coverage.

By comparison, ERA5 global reanalysis can provide a seamless coverage of

temperature and wind, and it is the last version of the European Centre for
Medium-Range Weather Forecasts (ECMWF) model and has 137 model levels
(Hersbach et al., 2020). It experiences a lot of improvements, including the
statistically significant improvement in short-range forecasts by the Aeolus satellite
(Rennie et al., 2021). Its predecessor, ERA-Intrim, was found in particular wind shear
a factor of 2–3 lower simulated based on high-resolution radiosondes (Houchi et al.,
2010). Moreover, results show that whatever the location and the geophysical
conditions considered, biases between ERA-Interim and balloon wind measurements
increase as a function of altitude (Duruisseau et al., 2017). Recent studies have
suggested that the structure and variability of the trade winds in the lower troposphere
are reasonably reproduced in the ERA5 reanalysis based on the EUREC4A field
campaign (Savazzi et al., 2022). However, the similar comparison between ERA5 and
high-resolution radiosonde across a near-global area has largely been undetermined.
The proper estimation of wind shear and Brunt-Väisälä frequency is essential for the
determination of $Ri$.

Thus, our objectives are to: (1) The performance of ERA5 (137 model levels) at

different heights and climate zones in estimating wind shear and KHI occurrence
frequency, comparing with a large high-resolution (10-m) radiosonde dataset spanning
years from 2017 to 2022. (2) Based on the validation and comparison results, we pose
a question: how to use ERA5 for KHI study? (3) The global climatology of KHI
occurrence based on versatile measurements and products. (4) The dynamic
environment (GWs and mean flow) of KHI. These works would be valuable for the
understanding of the global distribution of KHI, and furthermore, turbulence fraction.
To this end, this analysis is organized as follows. Section 2 shows the data and
methods used. Section 3 represents the climatological variation of KHI and its
comparison with radiosonde. Section 4 ends with a summary.


## 2 Data and methods

### 2.1 High-resolution radiosonde dataset

As described in Guo et al. (2021) and Zhang et al. (2022), a high vertical resolution radiosonde (HVRRS) dataset gained from several organizations was adopted, spanning January 2017 to October 2022, in a total of 5.8 years. The organizations include the China Meteorological Administration (CMA), the U.S National Oceanic and Atmospheric Administration (NOAA), the Global Climate Observing System (GCOS) Reference Upper-Air Network (GRUAN), the Centre for Environmental Data Analysis of the United Kingdom (CEDA), University of Wyoming, Deutscher Wetterdienst, and ECMWF. In total, 115 million radiosonde profiles from 434 radiosonde stations released at regular synoptic times of 0000 UTC and 1200 UTC were collected to determine the value of $Ri$. These profiles were sampled at 0.5 Hz or 1 Hz, corresponding to a vertical resolution of approximately 10 m or 5 m. Thus, all the profiles were evenly interpreted to 10 m resolution in vertical by applying a cubic spline interpolation. In addition, the sounding with the burst height lower than 10 km above ground level (a.g.l) was directly discarded for further study. Meteorological variables, including temperature and wind speed, were prepared for the $Ri$ estimation.

One of the shortage of radiosonde measurements is its inadequate concentration over the polar and ocean regions (Xia et al., 2021). The geographical distribution of total profile number of each radiosonde station is demonstrated in Figure S1 in Support Information. The released radiosoundings over Europe, the United States, and Australia have good geographical coverage and time duration. Over some islands of oceans (e.g., the Pacific Ocean) there are dozens of stations that can provide high-resolution measurement. Over the polar regions, there are around thirty stations.

### 2.2 ERA5 reanalysis and the collocation procedure

ERA5 is the latest version of ECMWF meteorological reanalysis, benefiting from





a decade of developments in model physics, core dynamics, and data assimilation
(Hersbach et al., 2020). The wind and temperature fields are modelled by the ERA5
reanalysis on a spatial resolution of 0.25° latitude/longitude and a temporal resolution
of 1 hour. The reanalysis has 137 model levels, giving a vertical resolution of
approximately 300 m in the middle and upper troposphere and 500 m in the lower
stratosphere. The vertical resolution of ERA5 is illustrated in Figure S2. Compared to
ERA5 reanalysis, the HVRRS is hard to provide global seamless observations. Thus,
the collocation procedure between reanalysis and HVRRS goes as follows: (1) the
matched grid of ERA5 reanalysis is the nearest neighbor of radiosonde station; (2) the
regular synoptic start time of radiosonde and reanalysis needs to keep exact the same;
(3) the pressure coordinate of reanalysis is converted into geometric altitude to match
with HVRRS. In addition, the standard deviations of orography (SDOR) and
near-surface wind speed at 10-m in ERA5 reanalysis are extracted.

The relative error between HVRRS-based and ERA5-based quantities is

estimated by the ratio of deviations between HVRRS and ERA5 derived quantities to
the HVRRS one.

**2.3 The occurrence frequency of KHI and its uncertainty**

The burst of KHI is characterized by the occurrence of the $Ri$ under a critical

value which is frequently taken as 1/4, and $Ri$ is formulated as:
$$\mathrm{Ri} = \bar{N}^2/\bar{S}^2 \qquad (1)$$
where N is the Brunt-Väisälä frequency, S is the vertical wind shear, and the
overbar denotes a moving average in 200-m step to eliminate the influence of
small-scale fluctuations, such as turbulence and small-scale waves. In this case, the
matching quantities that include $Ri$, wind shear, and the Brunt-Väisälä frequency
between radiosonde and ERA5 profiles are actually handled in averaged 200 m
intervals. The occurrence frequency of KHI is defined as the ratio of $Ri<1/4$ relative
to all $Ri$ calculations at a specified time period or height interval.

In Eq.(1), the length scale of overbar could potentially impact the value of $Ri$, and





eventually, the occurrence frequency of KHI. In addition, the critical value of *Ri* and
the vertical resolution of archived radiosonde could also cause the change in *Ri* values.
We resample the HVRRS data to 50 m and 100 m, and range the length scale of
overbar from 100 m to 500 m, to diagnose the uncertainties raised by the length scale
of segments and the vertical resolution of dataset. As indicated in Figure 1, under the
same length scale of overbar, a sparser vertical grid inevitably leads to a lower
occurrence frequency of KHI. For instance, as the length scale set to 100 m, the
occurrence frequency of *Ri*<1/4 at 0–2 km above sea level (a.s.l) decreases from 22%
when vertical resolution is equal to 10 m to 16% for a vertical resolution of 50 m.
Moreover, a longer length-scale of segment generally yields a smaller occurrence
frequency. For example, as the vertical resolution of radiosonde is equal to 10 m, the
occurrence frequency at 10–15 km decreases from 9% when the length scale of
segment equals 100 m to 1% when it equals 500 m. It is interesting to note that the
occurrence frequency under a vertical resolution of 50 m and a segment interval of
100 m is a bit larger than that under a vertical resolution of 10 m and a segment of
200 m, possibly implying the fact that a shorter segment interval could be expected
for a sparser vertical resolution.
**2.4 Gravity wave energy**
The GW energy is extracted based on the broad spectral method, according to
Wang and Geller (2003). In this method, the magnitude of measured zonal wind (*u*),
meridional wind (*v*), and temperature (*T*) consisting of background states ($u_0$, $v_0$ and
$T_0$) that are determined by applying a second-order polynomial fit (Chen et al., 2018;
Zhang et al., 2022) and perturbations. Therefore, total perturbations are derived as:

$$(u', v', T') = (\mathrm{u}, \mathrm{v}, \mathrm{T}) - (u_0, v_0, T_0) \qquad (2)$$

The perturbations could include measurement noises, KH waves, GWs, and
planetary waves. Only the perturbations with vertical wavelengths of 0.3–6.9 km are
considered as GWs (Wang and Geller, 2003). By applying this band-pass filter, the
average gravity-wave kinetic energy per unit mass (energy density) and the average



potential energy density can be expressed as:
$$E_k = \frac{1}{2}\left[\overline{u'^2} + \overline{v'^2}\right] \qquad (3)$$

$$E_p = \frac{1}{2}\frac{g^2\overline{\hat{T}'^2}}{N^2} \qquad (4)$$

where $g$ is the gravitational constant, $\hat{T}' = T'/\bar{T}$ the normalized perturbation
temperature, and the overbar indicates an averaging over the tropospheric segment,
which is chosen as 2–8.9 km for all regions, expect the polar region, and it is selected
as 2–7.4 km for the polar region (Wang and Geller, 2003). Eventually, the total GW
energy $E_t$ is the sum of $E_k$ and $E_p$.

## 3 Results and Discussions


### 3.1 Comparisons of wind shear between HVRRS and ERA5 reanalysis


The variations in vertical shear of horizontal wind speed and the squared

Brunt-väisälä frequency entirely determine the $Ri$ magnitude. Figure 2 provides an
overview of the spatial distribution of wind shear at heights of 0–2 km a.s.l and 10–15
km a.s.l obtained from the HVRRS and ERA5 reanalysis, explicitly representing the
variations of shear in the planetary boundary layer (PBL) and the upper troposphere,
respectively. The shear in the PBL regime estimated by ERA5 reanalysis demonstrates
a strong spatial variation, and it is largely dependent on underlying terrains and
latitudes (Fig.2a). For example, large values in the PBL regime can most likely be
observed along the coastline, which could be attributed to the prevailing sea-breeze
circulation. Large wind shear is common in regions where stability changes rapidly
(Grasmick and Geerts, 2020). As compared to the HVRRS, these shears are slightly
underestimated by approximately 4 m/s/km, mostly based on continental sounding
measurements. However, the oceanic shear is hard to be quantitatively assessed by a
large number of *in-situ* radiosonde stations, with this aspect likely being evaluated by
the ship-based radiosonde. Over the tropical oceans, Savazzi et al. (2022) found the
wind bias between EUREC[4]A field campaign and the ERA5 reanalysis varies greatly
from day to day, attributing to the bias in wind forecasting in the ERA5 reanalysis.





Nevertheless, a close association between averaged ERA5-retrieved shears and
HVRRS-determined shears can be noticed in terms of geospatial distribution, with a
correlation coefficient of 0.36.
It is noteworthy that shears in the ERA5 reanalysis at heights of 10–15 km a.s.l is
dramatically underestimated by around 8 m/s/km, especially at middle latitudes,
compared to the HVRRS. The underestimation could partly be due to the coarse
vertical resolution (around 300-m) in the ERA5 reanalysis in this height interval.
However, the spatial distribution of the ERA5 shear still exhibits a significant positive
correlation with the HVRRS shear, with a correlation coefficient of 0.35 (Fig.2d).
Following Houchi et al. (2010), the monthly shears over seven typical climate
zones are separately investigated, which are defined as follows: Northern
Hemisphere/Southern Hemisphere polar (70°–90°), Northern Hemisphere/Southern
Hemisphere midlatitude (40°–70°), Northern Hemisphere/Southern Hemisphere
subtropics (20°–40°), and tropics (20°S–20°N). Over the polar region in the
Northern/Southern Hemisphere, HVRRS-based shears are exceptionally strong in the
lower stratosphere compared to those in the troposphere (Fig.3a, g), which could be
attributed to the stratospheric polar jet. However, the similar altitude variation can
hardly be found in ERA5-based shears that are dramatically underestimated by around
16 m/s/km in the lower stratosphere (Fig.3h, n). The results in midlatitudes reach a
similar conclusion (Fig.3b, f, i, m). Over subtropical regions in the Northern/Southern
Hemisphere, HVRRS-based shears are consistent strong at heights of 16–21 km a.s.l,
just above the subtropical jet stream (Fig.3c, e). However, in the ERA5 reanalysis, the
region with consistently strong shears can be noticed at approximately 16 km a.s.l
(Fig.3j, l), which is about 3 km lower than that in the HVRRS. In the tropics, the
signature of quasi-biennial oscillation (QBO) can be identified in the lower
stratosphere (Fig.3d, k).
Overall, the ERA5-based shears are underestimated at almost all investigated
heights and over all climate zones, especially in the lower stratosphere. The
comparison between HVRRS-based and ERA5-based shears at three typical regimes
are tabulated in Table 1, representing the comparison result in the PBL region, the





middle and upper troposphere, and the lower stratosphere. These metrics highlight
that ERA5-based shears are underestimated by approximately 5 m/s/km, 7.5 m/s/km,
10 m/s/km at heights of 0–2 km, 10–15 km, and 20–25 km a.s.l, respectively, which
are roughly consistent with Houchi et al. (2010).
By comparison, the ERA5-acquired $N^2$ averaged from the surface to 30 km a.g.l
is reliably estimated over all climate zones, with a relative error of around 14%, as
illustrated in Figure S3. This finding indicates that the ERA5 reanalysis can properly
present the static stability of the background atmosphere, but it is not properly
coincident with radiosonde in terms of the small-scale variability of dynamical
structures. Due to a lack of global measurement of the fine-structure of the upper-air
wind, however, the accuracy of ERA5-resolved shears is hard to be globally validated.
**3.2 Occurrence frequency of $Ri$<1/4 in HVRRS and ERA5 reanalysis**
As a prominent example, the monthly occurrence frequency of $Ri$<1/4 over the
Corpus Christi station (27.77 ° N, −97.5 ° W) during years from January 2017 to
October 2022 is illustrated in Figure 4. As a result, the monthly occurrence rate of
$Ri$<1/4 in the PBL regime determined from HVRRS is lower than the ERA5-based
one, with mean values of around 10.6% and 16.9%, respectively. In the lowermost 2
km, the vertical resolution of ERA5 reanalysis is less than 200 m, and it is less than
the moving segment interval in Eq.(1). The high occurrence frequency in the PBL
regime could be likely related to the convective activity that leads to a negative $N^2$.
Especially during the daytime, PBL is well mixed due to strong turbulence induced by
uprising thermals (Song et al., 2018). In addition, an obvious seasonal cycle of
occurrence frequencies is revealed by HVRRS in the middle and upper troposphere
and has a maximal in the spring season (March–April–May), which is consistent with
the finding in Zhang et al. (2019). In the vicinity of jet streams, the occurrence
frequency of $Ri$<1/4 is generally enhanced by large wind shears. However, the ERA5
reanalysis is hard to provide such a seasonal cycle pattern, and it is significantly
underestimated by around 8%, which could be attributed to the underestimation in



wind shears. In the lower stratosphere, both the HVRRS and ERA5 reanalysis provide
a low estimation of occurrence frequencies, with a value of around 1%.
Furthermore, on a large spatial scale the occurrence frequency of $Ri<1/4$ retrieved
by ERA5 reanalysis is remarkably underestimated in the free atmosphere, as
compared to the HVRRS. The annual variation of the occurrence frequency of $Ri<1/4$
over seven climate zones at 0 to 30 km a.g.l indicated by HVRRS and ERA5
reanalysis is further demonstrated in Figure 5. It is clearly seen that the occurrence
frequency of $Ri<1/4$ provided by ERA5 reanalysis is underestimated in all months,
over all climate zones, possibly implying that, in the free atmosphere, the threshold
value of 1/4 in Eq.(1) is too small for the ERA5 reanalysis to capture the occurrence
of KHI.
Therefore, the question posed here is, what is the proper threshold value of $Ri$ in
predicting the occurrence of KHI when using the ERA5 reanalysis? The occurrence
frequency of $Ri<1/4$ indicated by the HVRRS, the ERA5-determined occurrence
frequencies produced by $Ri<0.25$, $Ri<0.5$, $Ri<1$, $Ri<1.5$, and $Ri<2$ at all heights up to
30 km a.s.l are demonstrated in Figure 6. It is notable that over all climate zones and
in the free atmosphere, occurrence frequencies of $Ri<0.25$ and $Ri<0.5$ obtained from
the ERA5 reanalysis are undervalued, but the frequencies of $Ri<1.5$ and $Ri<2$ are
generally overvalued. Among others, the occurrence frequency of $Ri<1$ gives a close
estimation both in magnitude and spatial variation compared to HVRRS over all
climate zones.
Furthermore, the correlation coefficients between HVRRS-determined KHI
occurrence frequency and the ERA5-determined frequencies indicated by different
threshold values of $Ri$ at height levels of 0 to 30 km are illustrated in Figure 7. It is
worth noting that, in the troposphere, the ERA5-based frequencies indicated by $Ri<1$,
$Ri<1.5$, and $Ri<2$ are highly positively correlated with those from the HVRRS, with a
correlation coefficient of around 0.6 over all climate zones. In the lower stratosphere,
however, these coefficients rapidly decline to 0.1, which can be explained by the low
occurrence frequency of KHI in this height regime.
Combined the findings in Figures 6 and 7, in the free troposphere, we can





conclude that the ERA5-determined occurrence frequency of $Ri<1$ is closest to the
frequency of $Ri<1/4$ based on the HVRRS. In the free atmosphere, KHI is the
dominant source for clear-air turbulence (CAT) that is a well-known hazard to
aviation. Therefore, the global characterization of KHI occurrence frequency in the
free atmosphere obtained from ERA5 reanalysis could be of importance for
understanding the spatial-temporal variation of CAT. In the following sections, the
occurrence frequency of KHI (hereinafter $OF$(KHI)) is based on $Ri<1$ in ERA5
reanalysis and $Ri<1/4$ in HVRRS, unless otherwise noted.
**3.3 The $OF$(KHI) climatology**
For a first hint the global distributions of $OF$(KHI) provided by the ERA5
reanalysis at 0–2 km a.s.l and 10–15 km a.s.l are displayed in Figure 8. $OF$(KHI) in
the PBL region is considerably spatially heterogeneous. Over subtropical oceans in
the Northern/Southern Hemisphere, the intense $OF$(KHI) can be noticed and has a
magnitude of around 50% (Fig.8a). In addition, over the Sahara Desert the $OF$(KHI)
reaches as high as 65%. Interestingly, the spatial variation in $OF$(KHI) keeps high
consistency with that of planetary boundary layer height (PBLH), as shown in Figure
S4. Usually, in the PBL regime, a deeper PBLH that represents more vigorous
convection activities can predict a higher $OF$(KHI). These findings suggest that, in the
PBL regime, the burst of KHI is likely closely associated with thermal convection due
to the heating of the ground. Similarly, at heights of 10–15 km a.s.l, intensive
$OF$(KHI) can be viewed over subtropic regions and has a value of around 10%
(Fig.8b), which is likely attributed to upper tropospheric jets.
In comparison, the spatial-temporal variability of $OF$(KHI) indicated by HVRRS
keeps high consistency with that of ERA5 reanalysis over all climate zones and at all
heights up to 30 km (Figure 9), especially in the free troposphere. Obvious seasonal
cycles can be detected over subtropics and midlatitude regions in the troposphere by
both the HVRRS and ERA5 reanalysis. However, the ERA5-based $OF$(KHI) can only
reflect the backbone of the cycles, and it is hard to quantify the detailed variation like





the HVRRS does. For regions without high-resolved wind and temperature
measurements, the ERA5 model product could be a good choice to represent the
thermodynamic instability of background atmosphere. Although ERA5-based
*OF*(KHI) is consistent with the HVRRS-based one from a global perspective, it is
generally underestimated over polar regions (Fig.9a, g, h, n).
Furthermore, the seasonal variation of *OF*(KHI) over seven climate zones
indicated by two datasets is shown in Figure 10. Over midlatitude and subtropics
regions, the *OF*(KHI) quickly decreases from around 40% in the PBL regime to
around 6% at around 3 km and then increases to around 8% at around 9 km, and
eventually, it decreases to around 2% in the lower stratosphere (Fig.10b,c,e,f). Over
tropic regions, a primary peak can be clearly noticed at around 13 km, with a
maximum of 12% for the HVRRS and 20% for the ERA5 reanalysis (Fig.10d, k).
Over polar regions, the tropospheric *OF*(KHI) is significantly lower than that over
other climate zones, with values ranging from around 4% at heights of 2–8 km to 1%
in the lower stratosphere (Fig.10a,g).
As well, the latitude-altitude variation of ERA5-reterived *OF*(KHI) is clearly
notable. In the free atmosphere the highest occurrences can be noticed at height
intervals of 8–15 km over tropical zones in all seasons, with magnitudes of around
30%. A poleward decrease pattern can be clearly detected in all seasons, with values
varying from 30% at low latitudes to around 5% at high latitude in the middle and
upper troposphere, which is consistent with the report in Zhang et al. (2022).
In Table 2, the mean *OF*(KHI) magnitudes over seven climate zones and at three
typical altitude regimes are listed. In the PBL, the ERA5-based *OF*(KHI) is about 20%
larger than that of the HVRRS-based one. In the middle and upper stratosphere, the
ERA5-based *OF*(KHI) is reasonably well estimated, except that it is overestimated by
around 5.8% over the tropics region. In addition, ERA5 underestimates *OF*(KHI) by
around 0.5% in the lower stratosphere.
According to Fig.8a, it seems that low-level *OF*(KHI) is dependent on underlying
terrains. Therefore, we investigate the association of low-level HVRRS-determined
*OF*(KHI) with the standard deviation of orography (SDOR). At heights of 1–2 km





a.g.l, the underlying terrain with a large SDOR generally corresponds to a high
*OF*(KHI), with a correlation coefficient between *OF*(KHI) and SDOR of 0.24. Then,
the coefficient decreases to 0.15 at 3–4 km a.g.l (Fig.11b), and eventually, it equals
0.14 at 5–6 km a.g.l (Fig.11c). These findings indicate that, over mountainous areas, a
high low-level *OF*(KHI) would be expected.

Moreover, it is quite evident from Fig.8b that the *OF*(KHI) is largely enhanced

over the tropical ocean associated with El Niño Southern Oscillation (ENSO) events.
The most of the enhanced *OF*(KHI) can be identified over the Niño 3 region (5 °N–5 °
S, 150 ° W–90 ° W), and the time-height cross section of *OF*(KHI) during years of
2000 to 2022 is illustrated in Figure 12. The *OF*(KHI) at height region of 6–13 km are
evidently large, with values of around 40%, which is about 20% larger than the
climatological mean value (Fig.9j). More specifically, *OF*(KHI) during time periods
of La Niño events is obviously stronger than that during the EI Niño periods. The
identification of ENSO events is based on Ren et al. (2018), Li et al. (2022), and Lv et
al. (2022). It is also worth recalling here that the wind shear does not exhibit such an
anomaly over the Niño 3 region (Fig.2c), implying that the *OF*(KHI) anomaly could
likely be attributed to the ENSO-related tropical convective heating in the upper
troposphere.
**3.4 The dynamical environment of KHI**

In the PBL, the raised KHI could be attributed to the interaction between complex

terrain and low-level wind and thermal convection. While in the free atmosphere
where the convection activity is weak, KHI is preferentially generated from strong
wind shear, which is closely associated with mean flows and wave activities.

We first evaluate the association of low-level *OF*(KHI) with near-surface wind

speed for the HVRRS station with a SDOR greater than 50 (Figure 13). It is probably
not surprising that the *OF*(KHI) is positively correlated with near-surface wind speed
at both heights of 1–2 km and 3–4 km a.g.l, with correlation coefficients of 0.09 and
0.04, respectively. These low coefficients could be attributed to too large samples.





Therefore, we infer that the interaction between near-surface winds and complex
terrains could increase the magnitude of low-level $OF$(KHI).

The propagation of GW could raise strong wind shear, and therefore generate

KHI. Thereby, we investigate the joint distribution of $OF$(KHI) with tropospheric GW
total energy and wind shear (Figure 14). The latitudinal variation of GW total energy
exhibits a double-peak structure, with two peaks at around 30° in the
Northern/Southern Hemisphere (Fig.14a). Overall, large $OF$(KHI) always
corresponds to strong GW activities and large wind shears, likely indicating that GW
activity is crucial for the occurrence of KHI.

In addition, the interaction between low-level wind and mountain barrier could be

a source of orographic GWs (Zhang et al., 2022). We take orographic GW dissipation
in ERA5 reanalysis, which is the accumulated conversion of kinetic energy in the
mean flow into heat over the whole atmospheric column, as an indicator of the
strength of orographic GWs. It is interesting to note that monthly averaged orographic
GW dissipation and ERA5-determined $OF$(KHI) at heights from ground up to 30 km
demonstrates a close association (Figure S5). For instance, in the middle troposphere,
they are positively associated over mountainous areas such as the Rocky Mountains
and the Alps Mountain, with correlation coefficients of around 0.5. These findings
also suggest that the propagation of orographic GWs could be a potential source for
KHI.

At jet heights (10–15 km a.g.l), a large shear is easily induced by strong wind

speed. Figure 15 demonstrates the joint distribution of $OF$(KHI) with wind speed and
wind shear. Similarly, large $OF$(KHI) can be easily found when the wind speed
exceeds 20 m/s and wind shear is larger than 20 m/s/km. However, it is clear that
large wind speed is not a necessary condition for KHI. The occurrence of KHI favors
the mean flow with a speed exceeding 20 m/s, but it does not ensure to happen for an
extremely large wind speed.

In a short conclusion, in the troposphere, the occurrence of KHI favors the

dynamical environment with strong orographic or non-orographic GW activities and
relatively large mean flow (>20 m/s).





## 4 Conclusion and remarks


The occurrence of KHI is potential crucial for many implications, such as aircraft,
mass transfer, and climate change, just name a few, but it is very hard to be globally
understood due to its fine structure. This analysis uses high-resolution model products
and radiosondes to globally characterize the distribution of KHI occurrence frequency
from the years 2017 to 2022.
Wind shears are considerably underestimated at almost all heights and over all
climate zones by the ERA5 reanalysis, compared to the HVRRS. It is noteworthy that
shears in the ERA5 reanalysis at heights of 10–15 km a.s.l is dramatically
underestimated by around 8 m/s/km, especially at middle latitudes. However, the
spatial distribution of the ERA5 shear exhibits a statistically significant positive
correlation with the HVRRS shear. The underestimation therefore influences the
performance of KHI analysis. As a result, the ERA5-determined occurrence frequency
of $Ri<1/4$ in the free tropospheric is significantly underestimated. In addition, it is
poorly correlated with HVRRS-determined ones at all heights and over all climate
zones.
Interestingly, the ERA5-determined occurrence frequency of $Ri<1$ is highly
consistent with the frequency of $Ri<1/4$ obtained from HVRRS, in terms of magnitude
and temporal variation. Rather than $Ri<1/4$, the threshold value of $Ri<1$ could be more
proper when using ERA5 reanalysis for KHI study, especially in the middle and upper
troposphere over midlatitude and subtropic regions in the Northern/Southern
Hemisphere.
The climatology of $OF$(KHI) exhibits significant seasonal cycles over polar,
midlatitude, and subtropic regions. A poleward decrease can be clearly identified in
the middle and upper troposphere. In addition, the low-level $OF$(KHI) is positively
sensitive to the standard deviations of orography. Moreover, it is immediately obvious
that the $OF$(KHI) in the middle and upper troposphere of the Niño 3 region is largely
enhanced by the tropical convective heating.



Over the mountainous area, the low-level $OF$(KHI) favors large near-surface
wind speed. In the free troposphere, the $OF$(KHI) favors intensive orographic or
non-orographic GW activities and relatively large mean flow (>20 m/s).
Those findings could be valuable for pointing out the performance of ERA5
reanalysis in terms of representing KHI occurrence frequency, as compared to a
near-global high-resolution radiosonde measurement. In addition, the spatial-temporal
variability of $OF$(KHI) over different climate zones from near-ground up to 30 km is
quantitatively characterized, which could provide new insights that increase our
understanding of the fine structure of upper air.

**Acknowledgement**
The authors would like to acknowledge the National Meteorological Information
Centre (NMIC) of CMA, NOAA, Deutscher Wetterdienst (Climate Data Center), U.K
Centre for Environmental Data Analysis (CEDA), GRUAN, ECMWF, and the
University of Wyoming for continuously collecting and generously providing
high-resolution radiosonde data.

**Financial support**
This study jointly supported by the National Natural Science Foundation of China
under grants 42205074, 42127805, and 62101203, and the Research Grants of
Huazhong Agricultural University under grants No. 2662021XXQD002 and
2662021JC008.



**Competing interests**

The contact author has declared that neither they nor their co-authors have any competing interests

**Data availability**

The dataset can be accessed at ECMWF (2022).

**Author contributions**

JZ conceptualized this study. JS carried out the analysis with comments from other co-authors. JZ wrote the original manuscript. WW, SZ, TY, WD provided useful suggestions for the study. All authors contributed to the improvement of paper.

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







**Table 1**. Comparisons of mean wind shears between HVRRS and ERA5 reanalysis at
heights of 0–2 km a.s.l (a), 10–15 km a.s.l (b), and 20–25 km a.s.l (c).

**(a) Wind shear at 0–2 km a.s.l (m/s/km)**

|  | Polar (NH) | Midlatitude (NH) | Subtropics (NH) | Tropics | Subtropics (SH) | Midlatitude (SH) | Polar (SH) |
|---|---|---|---|---|---|---|---|
| HVRRS | 12.67 | 12.94 | 12.30 | 10.57 | 13.03 | 14.16 | 15.01 |
| ERA5 | 7.45 | 7.68 | 7.78 | 5.4 | 8.44 | 9.67 | 8.42 |

**(b) Wind shear at 10–15 km a.s.l (m/s/km)**

|  | | | | | | | |
|---|---|---|---|---|---|---|---|
| HVRRS | 13.23 | 14.71 | 13.02 | 9.40 | 13.28 | 14.64 | 13.00 |
| ERA5 | 4.22 | 6.13 | 6.82 | 5.86 | 6.86 | 5.20 | 3.42 |

**(c) Wind shear at 20–25 km a.s.l (m/s/km)**

|  | | | | | | | |
|---|---|---|---|---|---|---|---|
| HVRRS | 15.12 | 15.74 | 15.41 | 16.76 | 16.69 | 16.12 | 17.15 |
| ERA5 | 2.87 | 3.52 | 4.06 | 5.27 | 3.99 | 3.36 | 2.92 |





















**Table 2**. Similar to Tab.1 but for the occurrence frequency of KHI. Note that the
occurrence of KHI is indicated by $Ri<1/4$ in radiosonde, but it is identified with $Ri<1$
in ERA5 reanalysis.

| (a) *OF*(KHI) at 0–2 km a.s.l (%) | | | | | | |
|---|---|---|---|---|---|---|
| | Polar (NH) | Midlatitude (NH) | Subtropics (NH) | Tropics | Subtropics (SH) | Midlatitude (SH) | Polar (SH) |
| HVRRS | 9.56 | 16.10 | 15.78 | 13.08 | 16.98 | 15.38 | 13.97 |
| ERA5 | 26.91 | 33.85 | 35.70 | 37.27 | 40.56 | 40.46 | 26.55 |
| **(b) *OF*(KHI) at 10–15 km a.s.l (%)** | | | | | | |
| HVRRS | 0.53 | 2.22 | 5.44 | 11.22 | 6.17 | 1.55 | 0.62 |
| ERA5 | 0.44 | 2.62 | 6.86 | 17.03 | 7.15 | 1.67 | 0.28 |
| **(c) *OF*(KHI) at 20–25 km a.s.l (%)** | | | | | | |
| HVRRS | 0.36 | 0.49 | 0.43 | 0.5 | 0.40 | 0.67 | 1.35 |
| ERA5 | 0.06 | 0.07 | 0.04 | 0.1 | 0.06 | 0.06 | 0.04 |





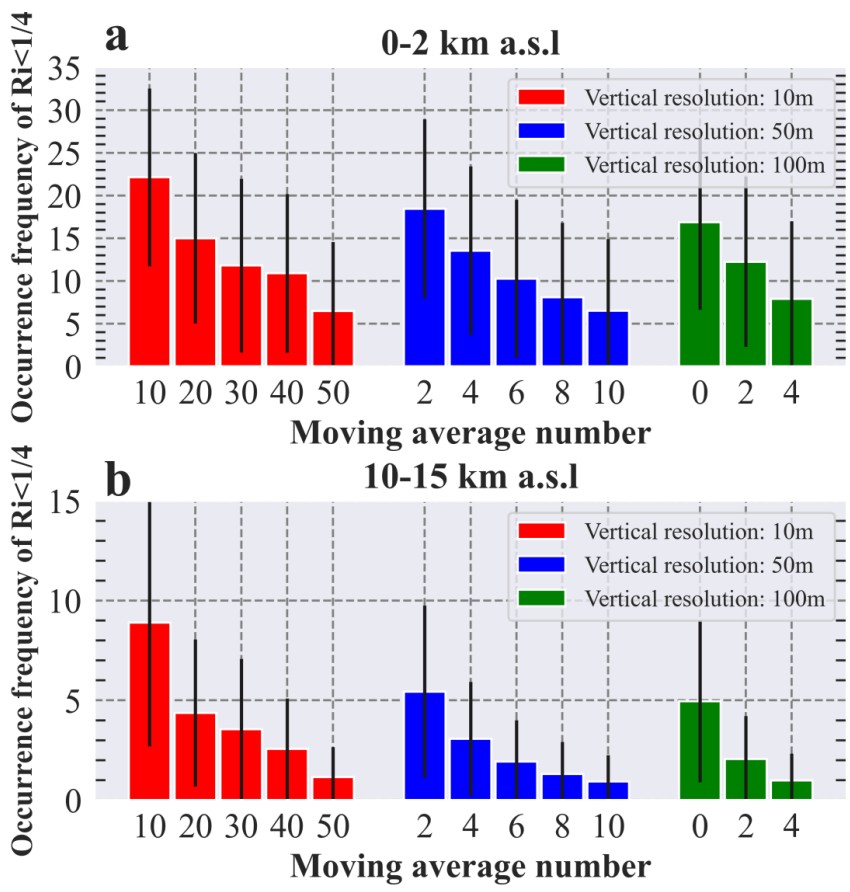

**Figure 1**. The averaged occurrence frequencies of $Ri<1/4$ at heights of 0–2 km a.s.l (a) and 10–15 km a.s.l (b), with vertical resolutions ranging 10-m to 100-m and moving point numbers increasing from 0 to 50. The error bars correspond to the standard deviation. The metrics are counted based on all radiosonde profiles during years 2017–2022.



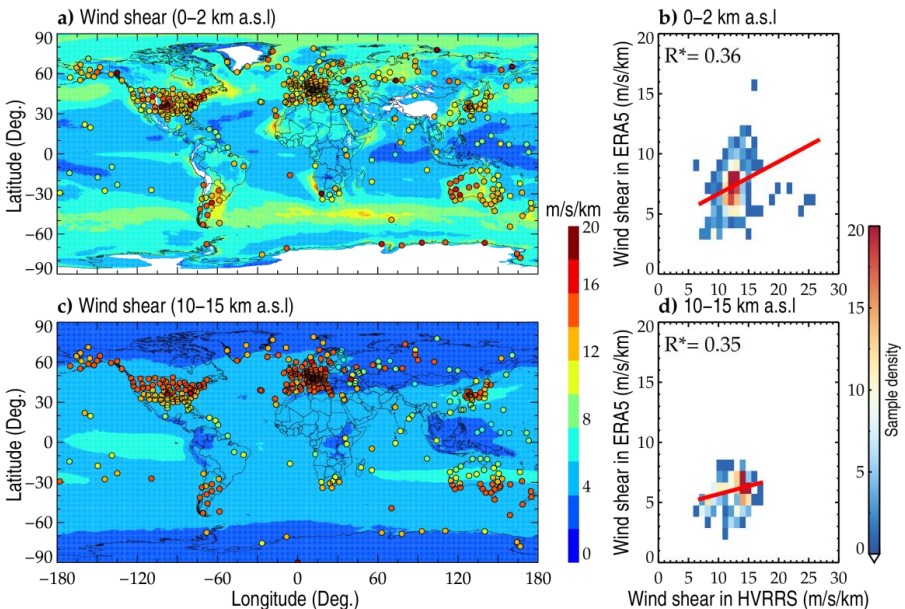

**Figure 2.** The spatial distribution of mean wind shear in ERA5 reanalysis at heights of 0–2 km a.s.l (a) and 10–15 km a.s.l (c), where the areas with a near-surface pressure lower than 800 hPa are masked with white. The overlaid colored circles represent the result in HVRRS at the same height levels. Each data point represents a vertically averaged value of the wind shear at one radiosonde station during the whole study period. Density plots (b, d) show the correlation between wind shears in HVRRS and ERA5 reanalysis. The ERA5 derived wind shears are spatially and temporally collocated with those of HVRRS. In addition, the red lines represent a least-squared linear regression, and the star superscripts indicate that values are statistically significant ($p<0.05$).



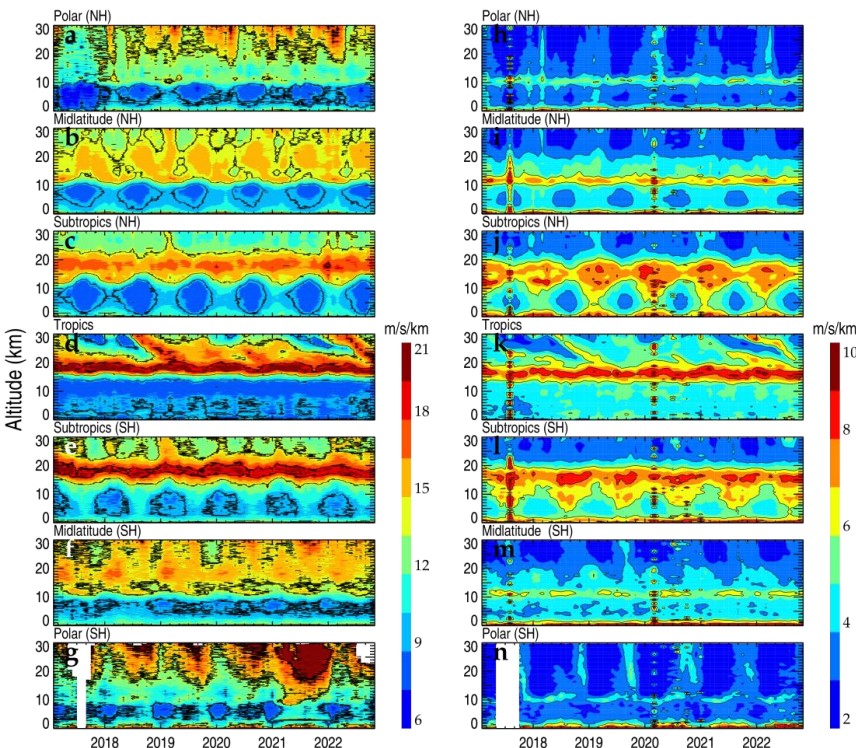

**Figure 3.** Monthly mean wind shears during years 2017–2022 in HVRRS (a–g) and ERA5 reanalysis (h–n) at different climate zones. The ERA5 derived wind shears are spatially and temporally collocated with those of HVRRS. NH=Northern Hemisphere; SH=Southern Hemisphere.



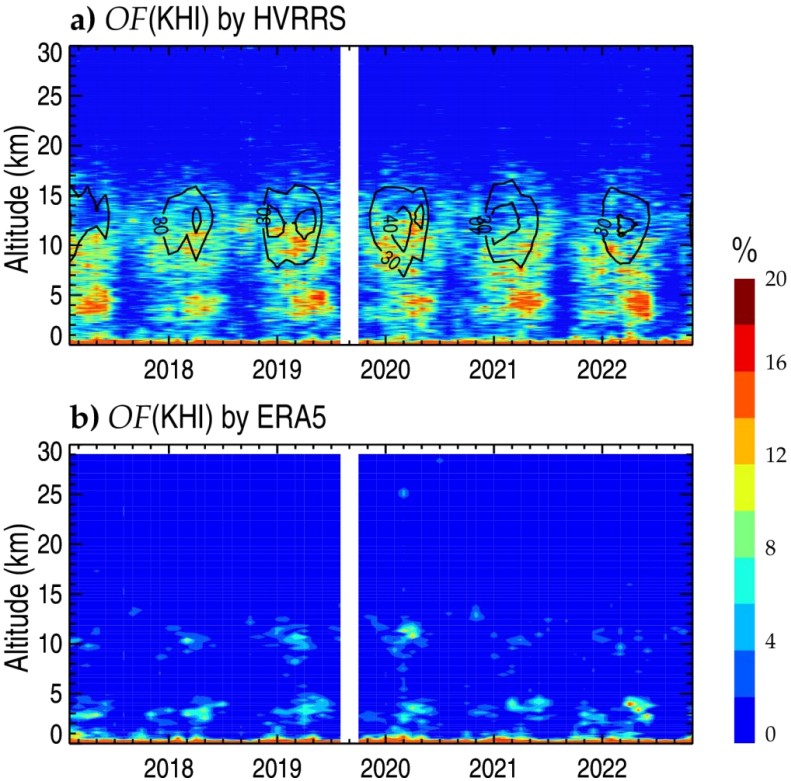

732

**Figure 4**. The monthly occurrence frequency of $Ri<1/4$ at Corpus Christi station (27.77 °N, −97.5 °W) in HVRRS (a) and ERA5 reanalysis (b). Note that the contour curves in (a) concern the mean horizontal wind speed, and that the ERA5 derived quantities are spatially and temporally collocated with those of HVRRS.

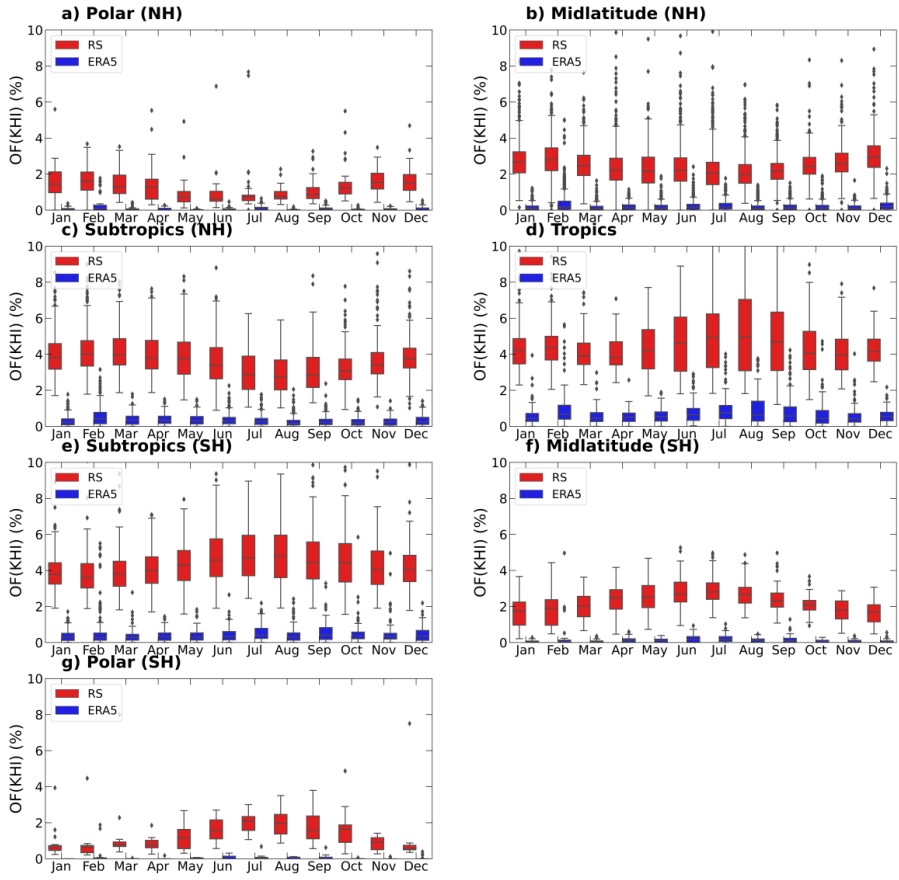

**Figure 5.** The annual cycles of the occurrence frequency of *Ri*<1/4 in different
climate zones at 10–15 km a.s.l. The red and blue boxes represent the frequencies in
HVRRS and ERA5 reanalysis, respectively. The ERA5 derived *Ri* is spatially and
temporally collocated with that of HVRRS. NH, Northern Hemisphere; SH, Southern
Hemisphere.



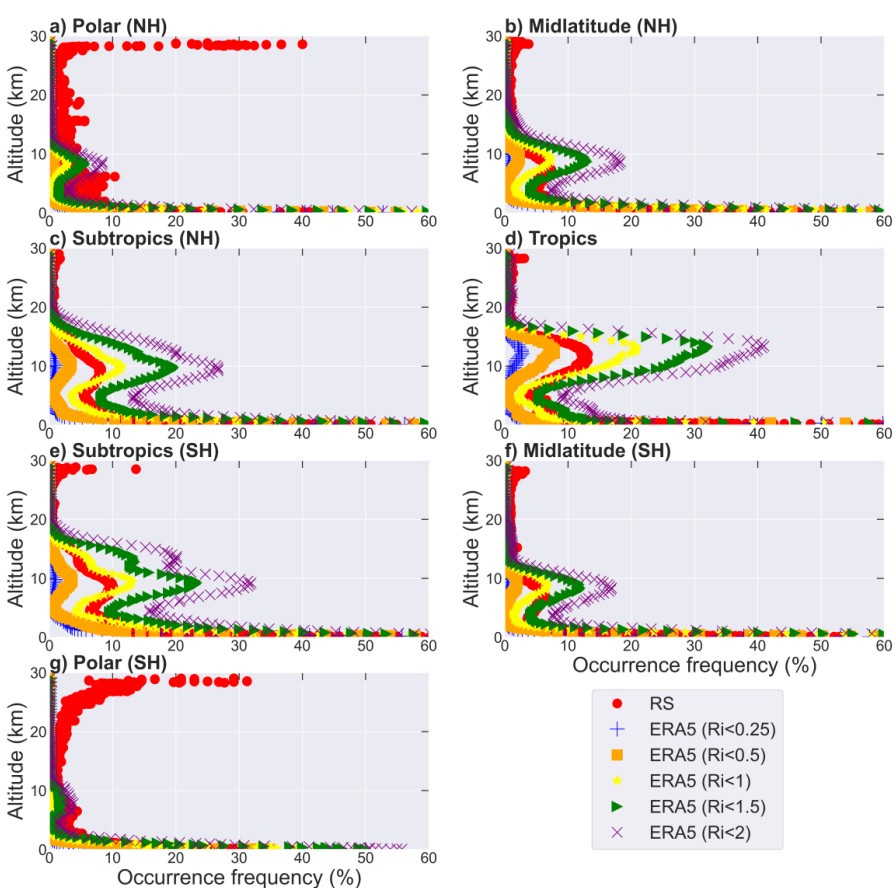

744

**Figure 6.** The altitude variation of the occurrence frequency of *Ri* below certain thresholds (0.25, 0.5, 1, 1.5, and 2) in ERA5 reanalysis in various climate zones. The ERA5 derived *Ri* is spatially and temporally collocated with that of HVRRS. The occurrences of *Ri*<1/4 in HVRRS are marked with red dots.

749

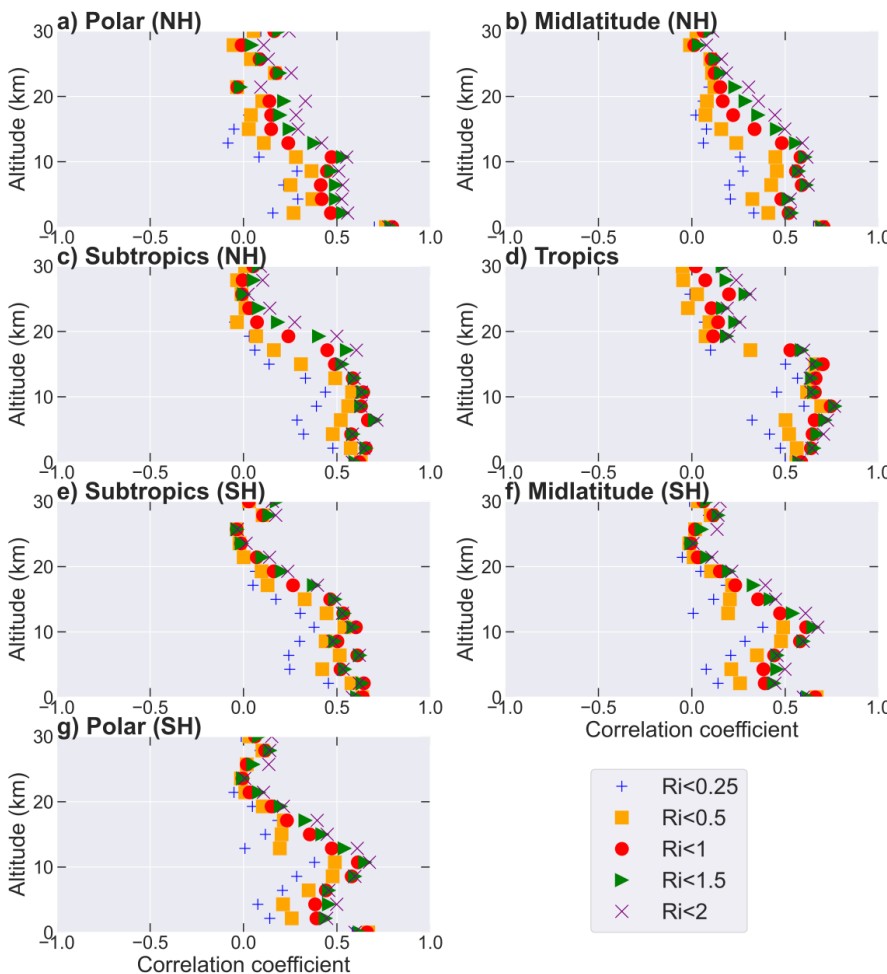

**Figure 7**. The correlation coefficients between monthly averaged KHI occurrence frequency in the HVRRS and the monthly occurrence frequency of $Ri$ below certain thresholds (0.25, 0.5, 1, 1.5, and 2) in ERA5 reanalysis. The ERA5 derived $Ri$ is spatially and temporally collocated with that of HVRRS. The coefficients in various climate zones are estimated in an increment of 2 km.



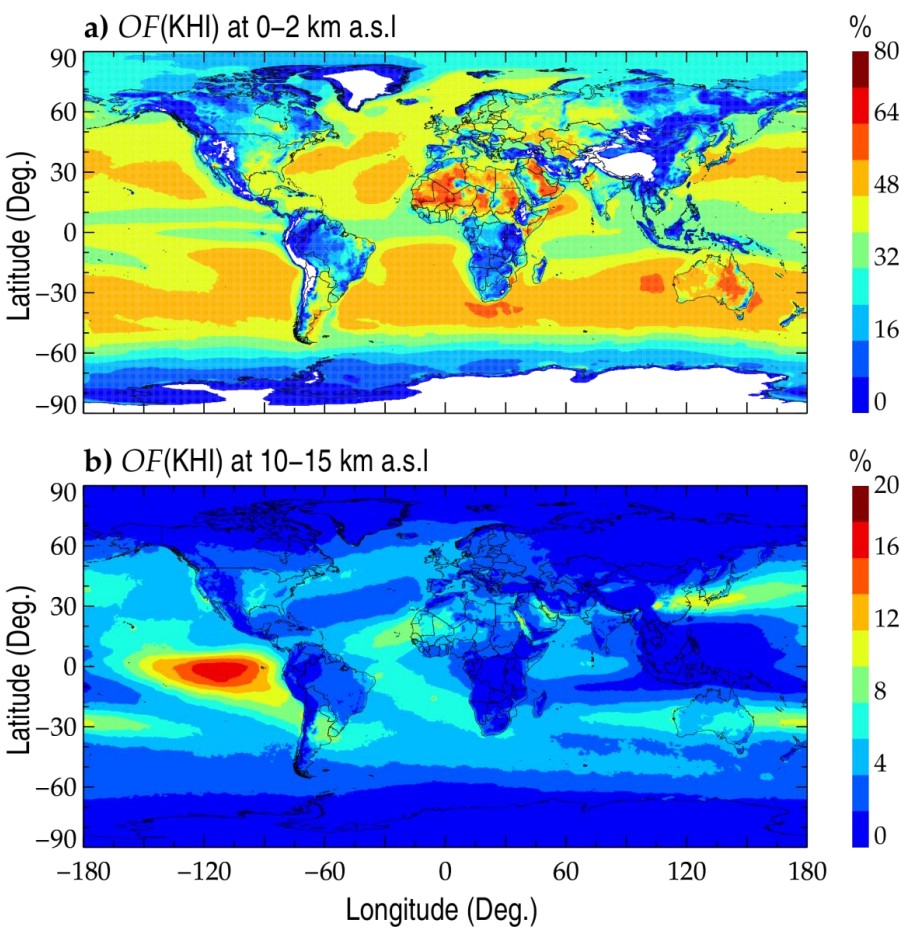

**Figure 8.** The spatial distribution of the mean occurrence frequency of KHI in ERA5 reanalysis at 0–2 km a.s.l (a) and 10–15 km a.s.l (b). Note that the threshold value of $Ri$ is set to 1.



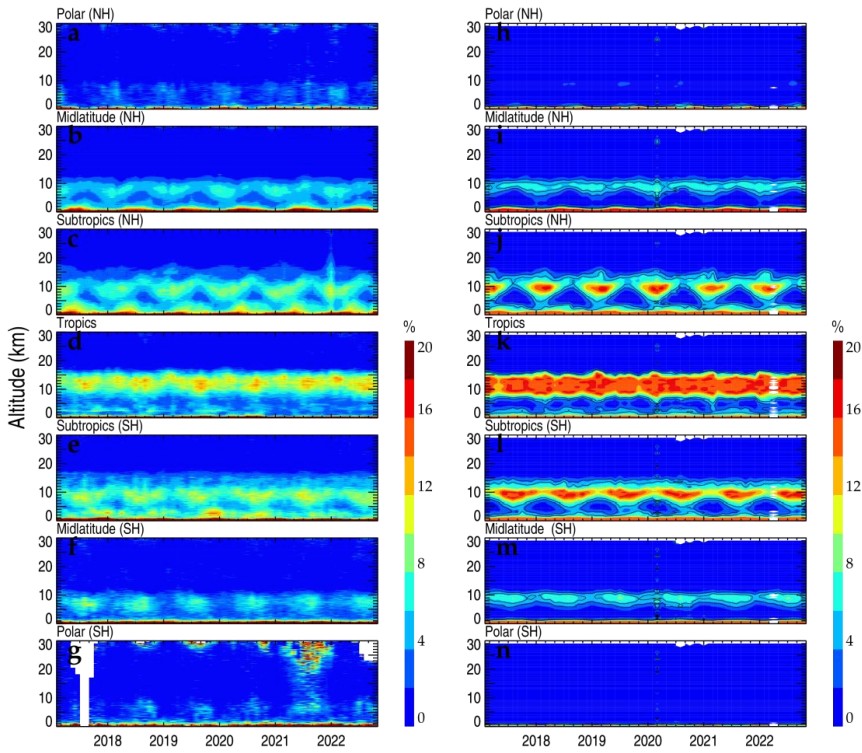

760

**Figure 9**. The monthly averaged $OF$(KHI) in the HVRRS (a–g) and ERA5 reanalysis
(h–n) in seven climate zones. NH=Northern Hemisphere; SH=Southern Hemisphere.



763

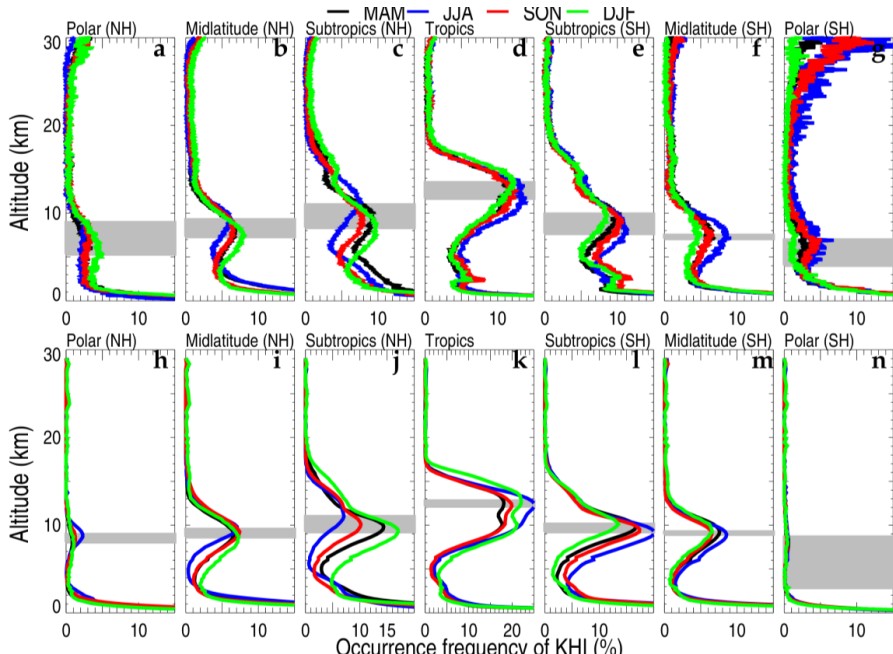

764

**Figure 10**. The seasonal averaged *OF*(KHI) in the HVRRS (a–g) and ERA5 reanalysis (h–m) in seven climate zones. The gray areas indicate the free tropospheric regime with maximal *OF*(KHI) in four seasons. MAM, March–April–May; JJA, June–July–August; SON, September–October–November; DJF, December–January–February. NH=Northern Hemisphere; SH=Southern Hemisphere.















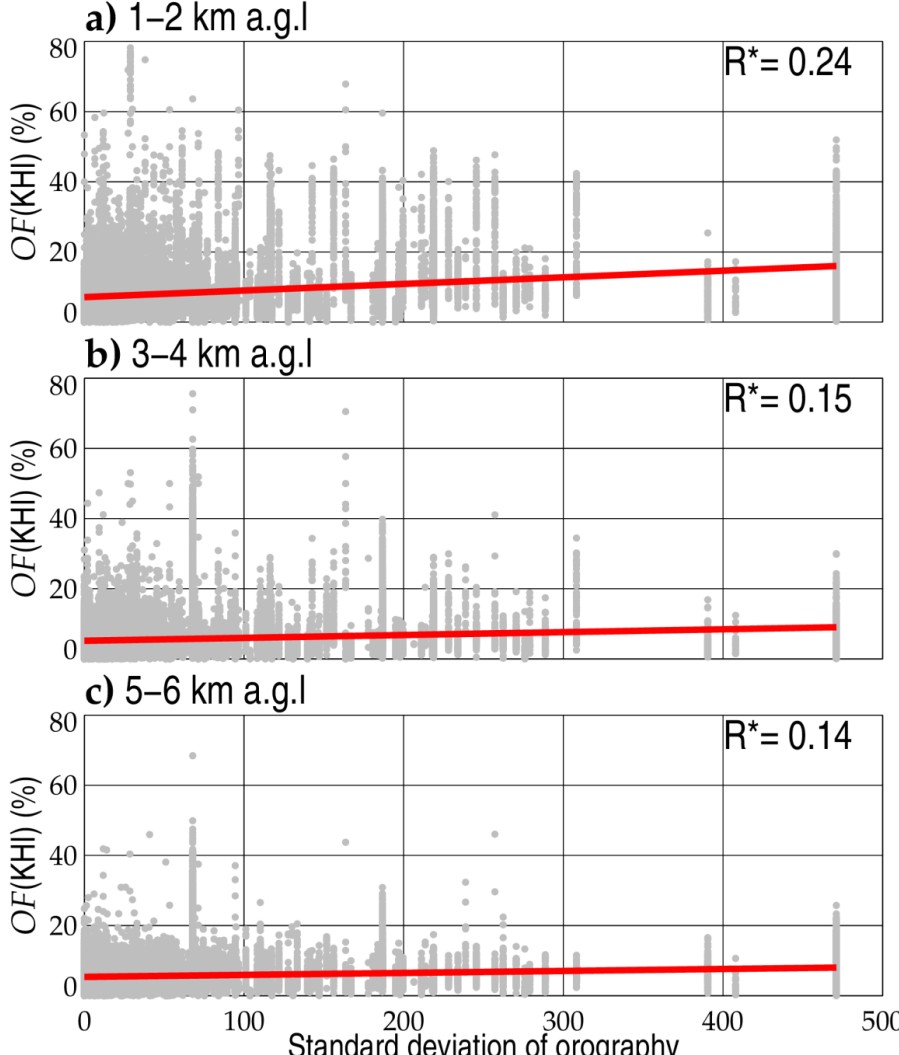


**Figure 11**. The association of HVRRS-determined *OF*(KHI) with different standard

deviations of orography. (a), (b), and (c) are for height ranges of 1–2 km, 3–4 km, and

5–6 km a.g.l, respectively. The correlation coefficients between *OF*(KHI) and

standard derivation of orography are marked in the top right corner, where the star

superscripts indicate that values are statistically significant ($p < 0.05$).





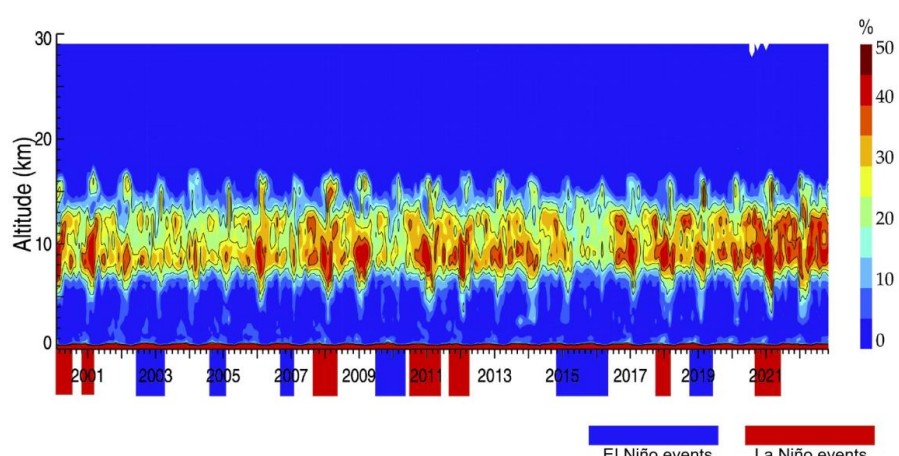


**Figure 12.** The monthly averaged $OF$(KHI) in ERA5 reanalysis over the Niño 3
region (5 °N–5 °S, 150 °W–90 °W). The blue and red shadings in time axis indicate
the time periods with EI Niño and La Niño events, respectively.

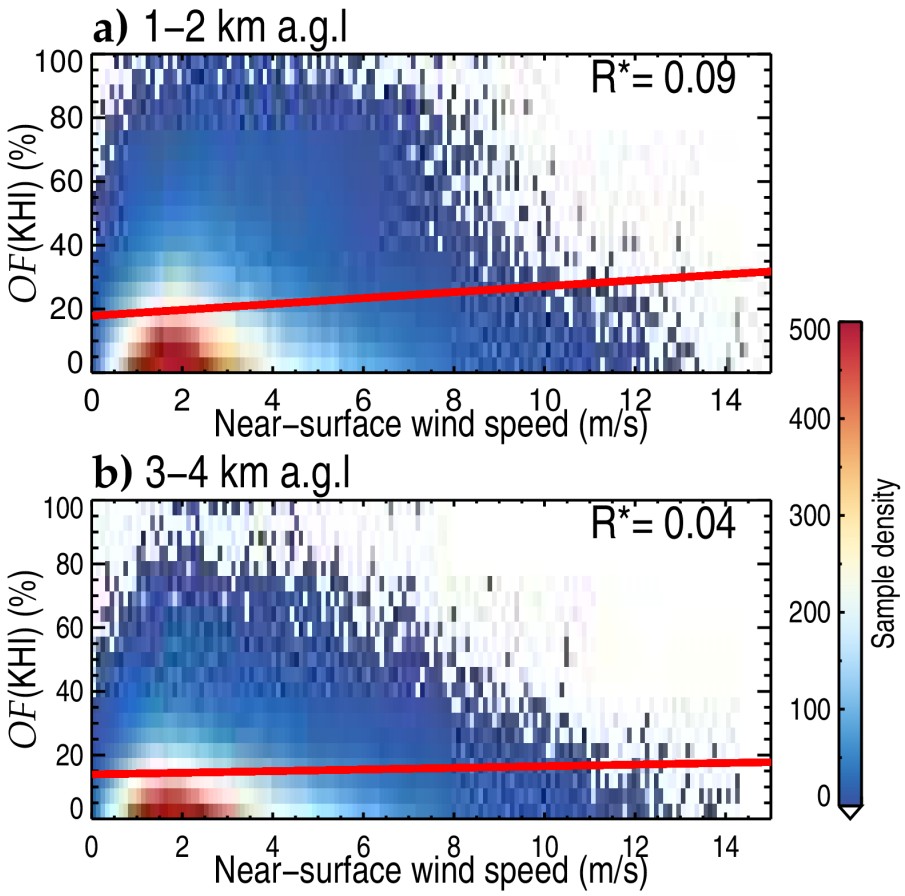


**Figure 13.** Density plots of HVRRS-derived *OF*(KHI) over terrain with standard
deviation of orography larger than 50 as a function of near-surface wind speed. The
red lines represent a least-squared linear regression. The star superscripts indicate that
values are statistically significant ($p < 0.05$).

799

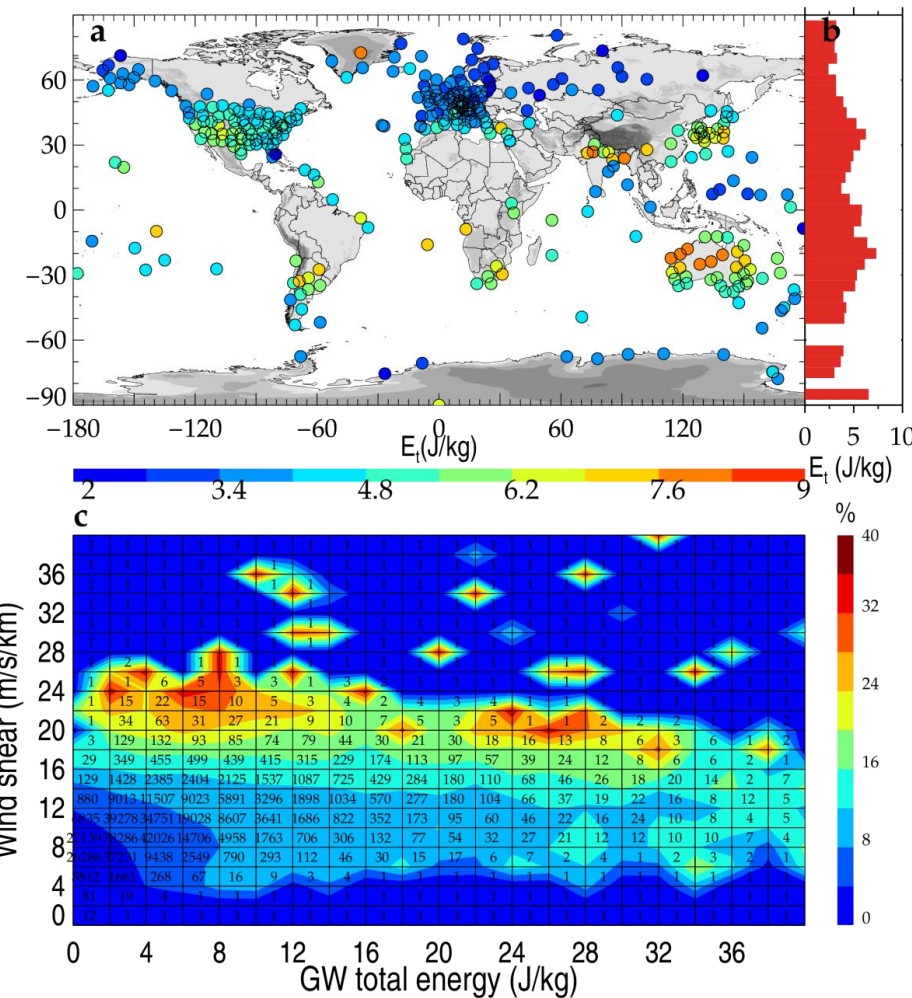

**Figure 14**. Geographical distribution of mean tropospheric GW total energy obtained from the HVRRS (a). The latitudinal variation of mean energy in a grid cell of 5° latitude (b). The joint distribution of *OF*(KHI) with GW energy and wind shear (c). The *OF*(KHI) and wind shear are derived from individual HVRRS profiles and vertically averaged over the tropospheric segment that is used for GW study. The numerical number in (c) indicates the matched profile number in each grid, using a bin size of 2 J/kg along the x axis and 2 m/s/km along the y axis.



811

812

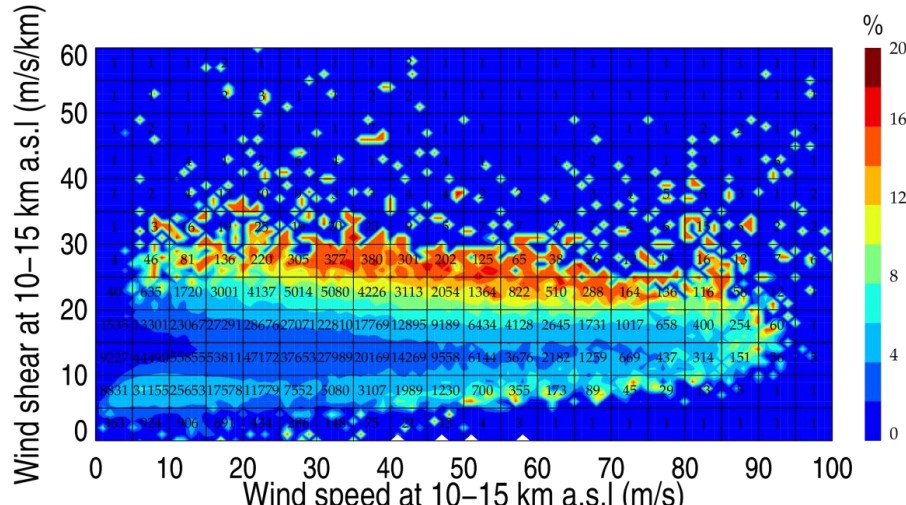

813

**Figure 15.** Joint distribution of HVRRS-derived wind speed, wind shear, and

*OF*(KHI), with a bin size of 5 m/s along the x axis and 5 m/s/km along the y axis.

Note that all the relationship is based on the mean result of individual profiles at

heights of 10–15 km a.g.l. The number indicates the matched profile number in each

grid.