# Peer review of "Occurrence frequency of Kelvin Helmholtz instability assessed by global high-resolution radiosonde and ERA5 reanalysis"

_Atmospheric Chemistry and Physics, 2022_

## Referee Comment (RC1)

The study analyses the occurrence frequency of subcritical Richardson numbers, in a comprehensive near-global multi-year HVRRS radiosonde data set, as well as in the co-located vertical profiles in the ECWMF ERA5 reanalysis. Based on several comparability criteria like the seasonality, vertical distribution, and correlation coefficients in different climate zones, an adjusted Richardson number threshold of $Ri_t=1$ is identified which exhibits characteristics most suited to diagnose regions in the ERA5 data which might be subjected to the occurrence of Kelvin Helmholtz instability.

I have two main concerns (see main comments below), one is more of a criticism on the wording, but it concerns the interpretation of the results at several points, and if the authors agree with the criticism, the wording and some aspects of the discussion throughout the manuscript have to be reworked.

My second main concern is that the interpretation of correlations and correlation coefficients in some of the analysis sections are too strong, which might therefore either need some reworking, a more detailed and broader description of the results leaving room for interpretation, or a reconsideration if these results are suited for publication.

Despite these main concerns, I found the results on the capability of the ERA5 to resolve subcritical Richardson numbers in comparison with the HVRRS interesting, and I think these along with the climatologies as key results could be suited for publication, however, only after consideration of the comments below.

I think the sections on the orographic and dynamical environment of regions of small Richardson numbers need clarification, mainly but not only in the context of the two main comments.

**Main comments:**

**01)**
I am uncertain to what amount the identification of vertical segments with $Ri<Ri_t$ (with $Ri_t=1/4$ in HVRRS and/or $Ri_t=1$ in ERA5) is equatable with the identification of KHI.

1. Subcritical Richardson numbers are a necessary but not sufficient criterion for the occurrence of dynamic instability and KHI.
2. Negative values for $N^2$ indicate convective instability. If the data was not filtered, I would expect convection to contribute significantly to the statistics of subcritical Richardson numbers.
   ◦ **HVRRS:**
     ▪ In Fig. 15, the secondary occurrence frequency maximum OF(Ri<1/4) below vertical wind shears of 10 m/s/km could be indicative of such convective contributions. But this is hard to say for sure because all values are vertical averages.
     ▪ At p.11 L.304 you include convection in the PBL in the interpretation.
   ◦ **ERA5:** Negative static stability in the model is (as I understand it) quickly resolved through the cloud parametrisation, and therefore rare in the model output. Still, regions of low static stability ($\rightarrow$ small Ri) could be indicative of regions prone to convection, particularly for the adjusted threshold of $Ri_t=1$.
     ▪ At p.13 L.361 you mention both KHI and convection in the PBL (diurnal cycle?).
     ▪ At p.15 L.418 you write the OF(KHI) maximum is related to convective heating.

I realize that Fritts et al. (2014) (who you cite in the introduction) discuss how the distinction between dynamic and convective instability can be *"challenging an misleading"*, but I am not sure how this discussion for Lidar measurements in the MLT region is applicable for the troposphere.

p.15 L.422 Would you describe convection in the troposphere generally as weak?

I would suggest to replace OF(KHI) in the manuscript by OF(Ri<Ri$_t$), and either expand the discussion and/or leave it to the readers interpretation which process dominates in each region and season. See also the more specific comments further below.

**02)**
I am not sure if the interpretation of the data based on the derived correlation coefficients is always justified:
1. The paragraph beginning at p.14 L.399: While I understand the idea behind the analysis and the approach, I am not sure that the interpretation is strongly supported by the data. From the plot it is not really possible to connect the scattered data with the derived linear correlation in Fig. 11. I would assume that a 2d-histogram would reveal that the highest data density is clustered at low SDOR, and that the moderate-to-large SDOR values are negligible for the estimation of the correlation coefficient. Therefore, I would say the interpretation *"over mountainous areas a high low-level OF(Ri<1) would be expected"* is too strong of a statement based on the data. But of course this is open for discussion.
   Furthermore, could you point out which regions in Fig. 8 you mean with the first sentence at p.14 L.399?
2. The paragraph beginning at p.15 L.425: Again, I understand the approach of the analysis and the idea behind it, but I would not agree that the conclusion is justified. The correlation is very weak. It is likely dominated by the point cloud at highest data density at low near-surface-winds. I would not agree that this supports the interpretation at the end of the paragraph.
3. For comparison: At p.17. L.473 you write: *"[The ERA5 determined occurrence frequency of Ri<1/4] is poorly correlated with HVRRS-determined ones at all heights and over all climate zones."*
   If this refers to Fig. 7, it might be too strong of a statement considering the overall interpretation of correlation coefficients in the manuscript.

**Other scientific comments:**

**ABSTRACT**

p.2 L.32 I would prefer to write *"KHI is indicated by the critical value of dimensionless Richardson (Ri) number, which is predicted to be ¼ from linear stability analysis."* or something along these lines.
Otherwise a reader without background knowledge might perceive the critical Richardson number as a purely empirical threshold.

p.2 L34 and L.36 I would suggest to delete the brackets with *"137 level"* and *"10 m"*, since these values are not directly comparable.

p.2 L.45 In the context of major comment 01, I would suggest to replace *"The occurrence frequency of KHI..."* with *"The occurrence frequency of subcritical Richardson numbers.."*

p.2 L45 From Fig. 9d and 10k I would argue that there is also a seasonality in the tropics. I see your point that it might overall be less pronounced, but I still would prefer to not exclude the tropics specifically. There are large scale flow features like the Summer Asian Monsoon and the tropical

easterly jet or the variability of the Walker circulation (also in relation to ENSO) which could be associated with the tropical seasonality in subcritical Richardson numbers (Roja Raman et al. (2009), Sunilkumar et al. (2015), Kaluza et al. (2021)).

**INTRODUCTION**

p.3 L.70 Baumgarten and Fritts (2014) focus on the MLT region, and although the statement (or parts of it) might be applicable to the region analyzed here, maybe there is a more general source in literature which describes the environment in which KHI arises.

p.3 L.73 Just to clarify, with *"the associated large gradient in jet stream"* you mean the wind gradients? Or stability gradients? Maybe you could rephrase it to make it more clear.

p.3 L.74 *"which may increase clear-air turbulence (Williams and Joshi, 2013)."*
Do you mean it could increase in a changing climate? If this is the case, maybe rephrase it in an individual sentence.

p.3 L.81 Maybe remove the sentence *"Among others, KHI is one of the most common causes of turbulence throughout the atmosphere from Earth's surface to the lower thermosphere (Fritts et al., 2011; Sharman et al., 2012)."* This is somewhat repeated from the sentence before.

p.4 L.89 *"Values of Ri close to zero favor strong instability, deep billows, and relatively intense turbulence, whereas values of Ri closer to ¼ favor weak instability, shallow billows (Fritts et al., 2011)."* I am not sure if this information is necessary for this study.

p.4 L.91 *"The threshold value of Ri can be potentially used an indicator of turbulence (for instance, Jaeger et al., 2007)."*
Suggestion for clarification: *"The Richardson number criterion can be applied as a turbulence diagnostic in numerical model output (e.g. Sharman and Pearson, 2017), and it has been used as such in climatological studies on the occurrence of clear air turbulence (Jaeger and Sprenger, 2007)."*
Since you cite Jaeger et al. (2007), a more direct comparison with your results in the later sections could be interesting, since Jaeger et al. also presents a Richardson number climatology with an adjusted $Ri_t$ threshold. And it would put your results in context with the literature.
Kunkel et al. (2019) include a brief discussion on the capability of ECMWF models to resolve subcritical Richardson numbers, and argue as well that $Ri_t=1$ might be a good proxy for observed KHI (although only in a case study).
A very recent study on this subject is by Lee at al. (2023), in their climatology on UTLS turbulence diagnostics. They also set the Richardson number threshold from 0-1.

p.4 L.101 *"While in numerical models, turbulent dissipation rate, turbulent diffusivity and other parameters representing turbulent mixing efficiency are the most basic parameters, which need to be accurately parameterized to evaluate the impact of turbulent effect on matter and energy distribution (Gavrilov et al., 2005). For this reason, the indices of turbulence, such as large wind shear, small Ri, the negative squared Brunt-väisälä frequency, could be a great tool to characterize turbulence (Jaeger et al., 2007)."*
I am not sure how the two sentences and the two citations fit together. Can you maybe rephrase the key statement you want to make here?

p.5 L.121 Is AEOLUS data assimilated in ERA5? At all, or consistently for the time period analyzed?

p.5 L.134 Suggestion: *"Thus, our objectives are to: (1) Evaluate the performance of ERA5 at different heights and climate zones in estimating wind shear and small Richardson number occurrence frequencies, in comparison with a large high-resolution radiosonde dataset spanning the years from 2017 to 2022."*

**2.1 HIGH-RESOLUTION RADIOSONDE DATA SET**

p.6 L.155 Are 115 million radiosonde profiles analyzed in this study? I was wondering because taking the upper limit of the color scale in Fig. S1 times the number of radiosonde stations 3000*434 equals about 1.3 million soundings. Maybe I am missing something.

**2.2 ERA5 REANALYSIS AND THE COLLOCATION PROCEDURE**

p.7 L.177 I would suggest to delete the part *"...and 500 m in the lower stratosphere."* It is not really necessary nor supported by Fig. S2.

p.7 L.184 If the orographic gravity wave dispersion is an ECMWF product, it should be listed here as well.

**2.3 THE OCCURRENCE FREQUENCY OF KHI AND ITS UNCERTAINTY**

p.7 L.190 I am missing an introduction of the critical Richardson number as a quantity which can be derived from linear theory. I personally would include this information in a study which focuses on the Richardson number.

p.7 L.193 please give the definition of the Brunt-Väisälä frequency and the vertical wind shear.

p.7 L.198 *"The occurrence frequency of KHI is defined as the ratio of Ri<1/4 relative to all Ri calculations at a specified time period or height interval."*
I don't think this sentence is necessary, and it can be misleading since you vary the threshold for the Richardson number.

p.7 L.200 *"In Eq.(1), the length scale of overbar could potentially impact the value of Ri, and eventually, the occurrence frequency of KHI. In addition, the critical value of Ri and the vertical resolution of archived radiosonde could also cause the change in Ri values."*
I would suggest to rephrase these two sentences to make the key point more clear.
Suggestion: *"The Richardson number calculated from Eq.(1) depends on the vertical resolution of the underlying data, as well as on the averaging interval. Ultimately, this influences the estimated occurrence frequency for subcritical Richardson numbers as a proxy for KHI. We resample the HVRRS data..."*

**2.4 GRAVITY WAVE ENERGY**

p.8 L.226 *"Only the perturbations with vertical wavelengths of 0.3–6.9 km are considered as GWs (Wang and Geller, 2003)."*

For clarification, is this due to the vertical limitation from 2-8.9 km (i.e. 6.9 km as an upper limit) and the effective vertical resolution of 150m (i.e., half a wavelength) in Wang and Geller (2003)?
Or do I misread this information?
How does this apply to your data?
Maybe this information should be provided after the description of the *"tropospheric segment"*.

**3 RESULTS AND DISCUSSION**

p.9 L.250 please provide a link to Table 1 if the value 4m/s/km is taken from this table.
If it is not the case, please indicate how the value was derived.

p.9 L.251 *"However, the oceanic shear is hard to be quantitatively assessed by a large number of in-situ radiosonde stations, with this aspect likely being evaluated by the ship-based radiosonde."*
I am not sure if this sentence is necessary, from the data description it is clear that radiosonde soundings over the oceans are sparse.

p.9 L.253 *"Over the tropical oceans, Savazzi et al. (2022) found the wind bias between EUREC4A field campaign and the ERA5 reanalysis varies greatly from day to day, attributing to the bias in wind forecasting in the ERA5 reanalysis."*
Again just a suggestion, but I personally would remove this sentence as well, since Savazzi et al. (2022) has been cited in the introduction, and the statement made here is not directly linked to the analysis. It might increase the reading flow if the manuscript is a bit more compact.

p.10 L.258 Maybe provide a link to Fig. 2b at the end of the sentence.

p.10 L.259 Suggestion: *"It is noteworthy that shear in the ERA5 reanalysis at heights of 10–15 km a.s.l. is significantly underestimated compared to the HVRRS, especially at middle latitudes, with a mean absolute error for all stations of about 8 m/s/km (Table 1).*
(Assuming the value 8m/s/km is taken from Table 1)

*p.10 L.265* Suggestion: *"Following Houchi et al. (2010), the monthly shears over seven typical climate zones are separately investigated (Fig. 3), which are defined as follows: polar (70°–90°), mid latitudes (40°–70°), subtropics (20°–40°), and tropics (20°S–20°N).*
*Over the polar region HVRRS-based shears are exceptionally strong in the stratosphere (Fig.3a, g), which could be attributed to the stratospheric polar jet."*
I think it is clear from the data description and the plots that both Hemispheres are analyzed separately.

p.10. L.274 Is the value 16 m/s/km taken from Table 1? If this is the case, please provide a link in the text. If it is not the case, please indicate how the value was derived.

p.10. L.275 Maybe remove *"in the Northern/Southern Hemisphere"*, it is again implied.

p.10 L.277 and Fig.3 h-n: Could you please comment on the vertically aligned wind shear structures in the ERA5 derived wind shear in summer 2017 and beginning of 2020? Are these real or artefacts due to the data processing? They look suspicious particularly since the figure shows monthly averages.
Furthermore, why do the regions of missing values in Fig. 3g and 3n differ (mid 2017 and end 2022)?

p. 10 L.279: *"…which is about 3 km lower than that in the HVRRS."*

Just out of interest, would you say this is the same shear layer shifted downward, i.e., a 3 km altitude bias in the dynamic structure of the UTLS?
Or are the shear magnitudes at the altitude of the maximum in ERA5 comparable to the ones in the HVRRS and the model fails to resolve the further increasing shear in the stratosphere?

p.10 L.283 I would suggest to remove the second part of the sentence, and just write: *"The comparison between HVRRS-based and ERA5-based shears at three typical regimes are tabulated in Table 1."*
Because the altitude range from 10-15 km is not the middle and upper troposphere, particularly at high latitudes.
Furthermore, if the mean absolute errors at page 9 and 10 (4m/s/km, 8m/s/km and 16m/s/km) refer to Table 1, then it would be better to introduce Table 1 beforehand.

p.11 L.292 (Fig. S3): Just a general thought, it could be interesting to see this comparison for the troposphere and the stratosphere separately, or for different altitude intervals.

p.11 L.304 *"The high occurrence frequency in the PBL regime could be likely related to the convective activity that leads to a negative $N^2$."*
Please see major comment 01.

p.11 L.307 *"In addition, an obvious seasonal cycle of occurrence frequencies is revealed by HVRRS in the middle and upper troposphere and has a maximal in the spring season (March–April–May), which is consistent with the finding in Zhang et al. (2019)."*
The maximum in the middle troposphere is during spring, the upper tropospheric maximum is also evident in winter I would say.

p.11 L.313 What does the value 8% refer to? Is it an estimation from Fig.4, or was it calculated from the data? If the latter is the case, which data exactly?

p.12 L.319 In Fig. 5 it says "10-15 km".

Figure 6: Maybe line plots would increase the readability compared to scatter plots. Or smaller markers.

p.13 L.345 Please see major comment 01. Of course this is open to discussion, but I would suggest to replace OF(KHI) in the following analyses with OF($Ri_{ERA5}$<1) or something similar. I don't think this lessens the impact of the results.

p.13 L.358 I am not sure if I can follow this argumentation. Could you point out exemplary regions for the spatial consistency?
How meaningful is it to compare multi-year averages for these two highly variable quantities?

p.13 L.361 See major comment 01.

p.13 L.366 Suggestion: *"In the free troposphere the spatial-temporal variability of OF($Ri_{ERA5}$<1) keeps high consistency with OF($Ri_{HVRRS}$<1/4) over all climate zones."*
I would refrain from specifying the altitude range up to 30km because of the stratospheric polar maxima in the HVRRS.

Figure 9:
- What are the vertically aligned structures in the ERA5 analysis at the beginning of 2020?
- Why is the ERA5 data here (and in Fig. 4) cropped at 29 km?

- Why are there missing data patches in the beginning of 2022 in the ERA5 data across all climate zones?
- Why are contour lines included in the ERA5 plots but not in the HVRRS? Maybe keep them similar for better comparison.

p.14 L.372 I would suggest to remove the sentence *"For regions without high-resolved wind and temperature measurements, the ERA5 model product could be a good choice to represent the thermodynamic instability of background atmosphere."*
This has been introduced in the motivation for the study.

p.14 L.377 Suggestion: *"The seasonal variation of OF(Ri<Ri$_t$) with Ri$_{t,HVRRS}$=1/4 and Ri$_{t,ERA5}$=1 for all climate zones is further analyzed in Figure 10. In the mid latitudes and subtropics, the OF(Ri<Ri$_t$) exhibits maximum values in the PBL, as well as a local minimum in the middle troposphere and a local maximum at altitudes around 9 km. In the stratosphere the occurrence frequencies decrease to values of the order of 1% (Fig.10b,c,e,f)."*
There is a significant variability in the vertical profiles across the different data sets, seasons and climate zones, and the single-value average occurrence frequencies in the current version of this paragraph might be confusing.

Figure 10: I would suggest to remove the gray shaded areas, I don't think they are necessary for the analysis.
Please adjust the x-axis range in Fig. 10k and 10l so that the JJA maxima are not cut off.

p.14 L.387 I would suggest to delete the whole paragraph down to line 392, I don't think it is important for the analysis, and kind of redundant with the paragraph before. I believe an overall reduction of the length would improve the manuscript.

p. 14. L.399 to L.406 Concerning the whole paragraph, please see major comment 02.

Figure 11: Could you plot the data as a binned 2d-histogram? What is the unit at the x-axis?

p.15 L.416 See major comment 01.

p.15 L.422 See major comment 01.

p.15 L.425 to L.431 Concerning the whole paragraph, please see major comment 02.
Provided that you agree with my criticism concerning the results of the analysis, I don't know what would be the best way for improvement:
You could either go without this part of the analysis (I don't think it would hurt the overall impact of the manuscript too much).
Or you could present the result without such a strong interpretation and leave more room for interpretation. I am not sure how meaningful this would be.
Or you could go into a more detailed analysis to try and sharpen the results. If this is within the scope of the study.

p.16 L.436 *"Overall, large OF(KHI) always corresponds to strong GW activities and large wind shears, likely indicating that GW activity is crucial for the occurrence of KHI."*
Is this derived from Fig. 14b? If so please link the Figure in the text.
I also think that a more detailed description would be helpful. What do you define as "strong GW activity" in the plot? Where do you see the correlation between strong GW activity and OF(Ri<Ri$_t$)?
Maybe I am misreading the plot, but I find it hard to follow the conclusion.

p.16 L.440 Is the orographic gravity wave dissipation an ECMWF product? If this is the case please include it in the data description.
And just for clarification, is this the quantity derived from the parametrised gravity wave drag due to subgrid-scale orography? So you identify regions and time steps (months) of strong resolved gravity wave activity based on the parametrised non-resolved gravity waves?

Figure S5: According to the text (p.16 L.443) the correlation is based on monthly averaged values (of the gravity wave dissipation and the $OF(Ri<Ri_t)$). Please include this information in the description of Fig. S5.
Again just for clarification: What years are the data basis for this plot? 2017-2022? So about 60 data points are correlated at each grid point?
So during months with strong parametrised gravity wave activity, a strong activity of resolved gravity waves can be expected, which then modify the flow and stability parameters of the resolved flow, and result in an enhanced occurrence frequency for low Richardson numbers.
However, in regions where according to Fig. 8b small Richardson numbers are rare (e.g. over the Rocky Mountains, the Andes, Scandinavia, the Alps).
It would be interesting to see a time series of the monthly averaged gravity wave dissipation and the $OF(Ri<Ri_t)$, for example over the Rocky Mountains. To get an impression of what this correlation means in terms of absolute values and the variability of $OF(Ri<Ri_t)$. But this is maybe beyond the scope of this study, since it is already pretty comprehensive.

Figure 15: I am not sure if I read this plot correctly. The bin size for the filled color contour (i.e., the $OF(Ri_{HVRRS}<1/4)$) is apparently something of the order of 1 m/s in the x-axis, and 1 m/s/km in the y-axis. If this is the case, shouldn't most of the filled contour display missing values, within 5x5 bins where only 1 or 2 matched profiles are located?

p.17 L.461 *"The occurrence of KHI is potential crucial for many implications, such as aircraft, mass transfer, and climate change, just name a few, but it is very hard to be globally understood due to its fine structure."*
Please rephrase this sentence.
Aircraft → aircraft safety?
Climate change → how?

p.17 L.463 Suggestion: *"This study uses the ERA5 as the latest reanalysis product from the ECMWF as well as a comprehensive data set of HVRRS radiosonde soundings to globally characterize the distribution of low Richardson numbers as a proxy for the occurrence of KHI, for the years 2017 to 2022."*

p.17 L.471 I would suggest to remove the sentence *"The underestimation therefore influences the performance of KHI analysis."*

p.17 L.473 See major comment 02.

p.17 L.479 *"...especially in the middle and upper troposphere over midlatitude and subtropic regions in the Northern/Southern Hemisphere."*
Why especially in these regions?

p.17 L.482 Suggestion: *"The climatology of $OF(Ri<Ri_t)$ exhibits significant seasonal cycles over all latitudes."*
When looking at Fig. 9 and 10 I don't see why the tropics should be specifically excluded here.

p.17 L.484 to p.18 L.490

This paragraph might need to be reworked based on how the manuscript was changed after the review.

p.18 L.491 Maybe this is a personal preference, but I would not write the final paragraph in subjunctive.

For example (again just a suggestion): *"Those findings are valuable for pointing out the performance of the ERA5 reanalysis in terms of resolving low Richardson numbers as a proxy for KHI, in comparison with a near-global high-resolution radiosonde measurement."*
Same for the last sentence.

p.19 L.516 Please be more specific and include both data sets (ERA5 and HVRRS).

**Technical corrections (typos etc.):**

Throughout the manuscript: Maybe it should say *"a.s.l."* instead of *"a.s.l"*?

p.3 L.67 Remove the sentence *"In addition, GW breaking has been identified as important sources of instability"*, this is already stated one sentence earlier.

p.4 L.108 I would suggest to either write *"The Richardson number is estimated by the…"*
or *"Ri is estimated by the..."*

p.6 L.167 *"...in the supporting information."*

p.7 L.179 Suggestion: *"Compared to ERA5, the HVRRS does not provide global seamless observations."*

p.9 L.248 . *"Large wind shear is common in regions where stability changes rapidly (Grasmick and Geerts, 2020)."* I would suggest to delete the sentence as it is already in the introduction.

p.10. L.259 *"shear"* instead of *"shears"*

p.11 L304 *"maximum"* instead of *"maximal"*

p.11 L.311 Suggestion: *"However, the ERA5 reanalysis does not provide such a seasonal cycle pattern.."*

p.12 L.331 *"underestimated"* instead of *"undervalued"*. Same for *"overvalued"*.

p.12 L.332 remove *"Among others"*

p.13 L.371 I am not sure about the word *"backbone"*. Maybe replace it with *"large scale structure"* or something along those lines?

p.15 L.408 Replace *"associated with El Niño Southern Oscillation (ENSO) events."* with *"associated with the El Niño Southern Oscillation (ENSO)."*

p.15 L.414 typo: *La Niña*

p.17 L.468 *"vertical wind shear"* instead of *"shears"*

**References:**

*Dutton, J. A., & Panofsky, H. A. (1970). Clear air turbulence: A mystery may be unfolding. Science, 167(3920), 937–944. https://doi.org/10.1126/science.167.3920.937*

*Kaluza, T., Kunkel, D., & Hoor, P. On the occurrence of strong vertical wind shear in the tropopause region: A 10-year ERA5 northern hemispheric study. Weather and Climate Dynamics, 2(3), 631–651. https://doi.org/10.5194/wcd-2-631-2021, 2021*

*Lee, J. H., Kim, J.-H., Sharman, R. D., Kim, J., & Son, S.-W. Climatology of Clear-Air Turbulence in upper troposphere and lower stratosphere in the Northern Hemisphere using ERA5 reanalysis data. Journal of Geophysical Research: Atmospheres, 128, e2022JD037679. https://doi.org/10.1029/2022JD037679, 2023*

*Roja Raman, M., Jagannadha Rao, V. V., Venkat Ratnam, M., Rajeevan, M., Rao, S. V., Narayana Rao, D., and Prabhakara Rao, N.: Characteristics of the Tropical Easterly Jet: Long-term trends and their features during active and break monsoon phases, J. Geophys. Res.-Atmos., 114, 1–14, https://doi.org/10.1029/2009JD012065, 2009.*

*Sharman, R. D., & Pearson, J. M. Prediction of energy dissipation rates for aviation turbulence. Part I: Forecasting nonconvective turbulence. Journal of Applied Meteorology and Climatology, 56(2), 317–337. https://doi.org/10.1175/JAMC-D-16-0205.1, 2017*

Sunilkumar, S. V., Muhsin, M., Parameswaran, K., Venkat Ratnam, M., Ramkumar, G., Rajeev, K., Krishna Murthy, B. V., Sambhu Namboodiri, K. V., Subrahmanyam, K. V., Kishore Kumar, K., and Shankar Das, S.: Characteristics of turbulence in the troposphere and lower stratosphere over the Indian Peninsula, J. Atmos. Sol.-Terr. Phys., 133, 36–53, https://doi.org/10.1016/j.jastp.2015.07.015, 2015.

---

## Referee Comment (RC2)

Review report on the paper **"Occurrence frequency of Kelvin Helmholtz instability assessed by global high-resolution radiosonde and ERA5 reanalysis"** by Shao et al. submitted to the journal Amospheric Chemistry and Physics.

**Overview**

This paper describes the spatial and temporal variability of instabilities in the atmosphere from a huge radiosonde (RS) database (115 million profiles (?) from 434 stations!). The resulting statistics from the radiosonde analysis are compared to those obtained from ERA5 reanalyses. The authors observe a severe underestimation of the vertical shear from ERA5. They find a better agreement between the instability climatology from RS and ERA5 by taking a threshold Ri < 1 when using ERA5.  The authors also observe a positive correlation with the standard deviation of the orography.

The analysis of the RS and ERA5 data is interesting and the climatological results seem very convincing. The comparison RS/ERA5 is also interesting. For these reasons, this work deserves publication. However, the interpretation of the results does not always seem to me to be correct. For instance, the fact of interpreting all instabilities detected from the Ri < 1/4 criterion as the result of Kelvin-Helmholtz instabilities is not justified, in particular in the boundary layer, or in the tropical troposphere.  Therefore, substantial modifications seem to me necessary, if only the title.

I therefore recommend that this article be published with some substantial modifications.

**Major comments**

**1) lines 190-21**7: about the method for estimating the gradients from the HVRRS profiles. If I understand correctly, a moving average of the estimated gradients is performed over height segments of 200 m. But over what vertical scale are the gradients evaluated (which are then averaged)? Are they calculated over 10 m differences? If so, why this choice if one purpose is to compare with the estimates from ERA5? Why not estimate the vertical gradients of HVRRS on vertical scales comparable to the model (100-400 m) for comparison? Moreover, it would be simple to adapt the resolution of the HVRRS gradients to the resolution of the model according to the altitude domains considered.

Figure 1 is illuminating on that purpose in showing that the estimates of the occurrence frequency (OF) of Ri on a 100 m scale, without averaging, is little different from that estimated on 10 m differences averaged over 200 m.

Such a resolution (10 m averaged on 200 m segments) is relevant to establish the climatology of instability occurrences (Ri < 1/4) obtained from HVRRS, but is arguably questionable for comparison to the climatology deduced from ERA5.

**2) paragraph 3.2**: The authors correctly note that the frequency of occurrence (OF) of KHIs depends on the vertical resolution of the gradients. The fact that the OFs from the model coincide better with those from the HVRRS with a threshold Ri < 1 is therefore fortuitous since it depends on the resolution of the HVRRS. (If you had calculated the gradients on a 50 m scale, and not 10 m, you would have lower OFs, and therefore a different threshold to apply on the model estimates). Can you please comment on this fact?

**3)** The authors systematically attribute Ri < 1/4 occurrences to KH instabilities. This is certainly not always the case. Thus, the diurnal boundary layer is very probably close to a neutral static stability (Ri ~ 0) without having anything to do with KH instabilities. The same is true in the tropical troposphere, where deep convective cells develops up to about 15 km altitude. The Ri < 1/4 occurrences are most likely a signature of an unstable flow, but not that this instability is due to a KHI. I recommend that the authors modify the discussion and conclusion paragraphs accordingly. As well as the title of the article.

4) Horizontal winds measured under radiosonde at the scale of a few tens of meters are affected by the chaotic movements of the gondola due to the pendulum and to the balloon's own movements (see for example Ingleby et al., 2022, https://doi.org/10.5194/amt-15-165-2022 and references therein). A low-pass filter is applied to the HVRRS profiles to reduce these effects. This filtering should have an impact on the effective resolution of the wind measurements (which is much larger than 10 m). Although it is difficult to assess the impact of this filtering, I suggest that you discuss this fact.

**Specific comments**

- **line 155**: 115 million HVRRS profiles??? Do you confirm?

- **Line 233**: a "tropospheric segment" from 2 to 8.9 km is chosen. Why this choice (if interested in the OF(KHI) in the 0-2 and 10-15 km height ranges?

- **line 240** (Fig. 2): what is the vertical resolution of the shear estimates from HVRRS? Please, specify.

- **lines 259 & 282**: Are the ERA5 shear estimates dramatically lower than the HVRRS estimates if the RS gradients are estimated with the same resolution as the model (i.e. 300 m for the 10-15 km altitude domain)?

- **line 306**: I agree with the statement that PBL is mixed by convection during daytime. Clearly, the occurrence of Ri < 1/4 is not attributable to KH instabilities in such cases.

- **line 376**: this statement is not visible in figure 9.

- **lines 381-3**: I doubt that the Ri < 1/4 occurrences are only due to KHI in this region where deep convective cells are frequent.

- **lines 414 & 793**: la Niño → la Niña

- line 436: I do not understand your concluding sentence, figure 14b showing precisely that the probability of occurrence Ri < 1/4 depends almost not on the total energy of the gravity waves, but almost exclusively on the horizontal wind shear (if it exceeds 18 m/s/km).

- **line 443-456**: I do not understand your conclusion about the wind speed. Figure 15 clearly shows that the occurrence Ri < 1/4 does not depend on the wind speed (if the wind speed exceeds a few m/s), but occurs with high probability if the shear exceeds ~20 m/s/km. This is a convincing result!

---

## Author Comment (AC1)

**Response to Reviewers #1 Comments**

We thank the associate editor, editor and two anonymous reviewers for their thoughtful and exhaustive comments and suggestions, which significantly help us to improve the quality of the manuscript. In this revised manuscript, we have revised the manuscript accordingly. Below, we indicate the original comment of the respective reviewer in blue and our point-to-point response is denoted in black.

Before addressing the comments, we would like to express our sincere gratitude to the reviewers for their exceptionally informative, constructive, and detailed comments.

**Reviewer #1 Evaluations:**

The study analyses the occurrence frequency of subcritical Richardson numbers, in a comprehensive near-global multi-year HVRRS radiosonde data set, as well as in the co-located vertical profiles in the ECWMF ERA5 reanalysis. Based on several comparability criteria like the seasonality, vertical distribution, and correlation coefficients in different climate zones, an adjusted Richardson number threshold of Rit=1 is identified which exhibits characteristics most suited to diagnose regions in the ERA5 data which might be subjected to the occurrence of Kelvin Helmholtz instability. I have two main concerns (see main comments below), one is more of a criticism on the wording, but it concerns the interpretation of the results at several points, and if the authors agree with the criticism, the wording and some aspects of the discussion throughout the manuscript have to be reworked.

My second main concern is that the interpretation of correlations and correlation coefficients in some of the analysis sections are too strong, which might therefore either need some reworking, a more detailed and broader description of the results leaving room for interpretation, or a reconsideration if these results are suited for publication.

Despite these main concerns, I found the results on the capability of the ERA5 to resolve subcritical Richardson numbers in comparison with the HVRRS interesting, and I think these along with the climatologies as key results could be suited for publication, however, only after consideration of the comments below.

I think the sections on the orographic and dynamical environment of regions of small Richardson numbers need clarification, mainly but not only in the context of the two main comments.

Response: We sincerely appreciate the reviewers for their thorough and insightful comments and assessments, which have greatly benefited our present work as well as our future research. We learned a lot from your suggestive comments. In the following,

we address your comments in detail, and we have revised some parts of our work based on your constructive suggestions, aiming to achieve your desired outcomes.

**Main comments:**

01)

I am uncertain to what amount the identification of vertical segments with Ri<Rit (with Rit=1/4 in HVRRS and/or Rit=1 in ERA5) is equatable with the identification of KHI.

Response: Thanks for the comments. In the updated file, we have stated this conclusion in the Conclusion and Remarks session:

"…In other words, under a similar occurrence frequency, the identification of vertical segments with $Ri<1$ in ERA5 is equitable with identification of vertical segments with $Ri<1/4$ using HVRRS…"

1.  Subcritical Richardson numbers are a necessary but not sufficient criterion for the occurrence of dynamic instability and KHI.

Response: We agree. In the Introduction section of the original manuscript, we also stated that Ri is not a good guide to instability character in general, and Ri>1/4 does not assure flow stability for superpositions of mean and GW motions. Despite these caveats, Ri<1/4 does provide a reasonable guide to expected local KHI structure in cases where clear KH billows arise.

In addition, in the revised manuscript, we principally emphasize the occurrence frequency of low Richardson number based on your professional and detailed comments.

2.  Negative values for N² indicate convective instability. If the data was not filtered, I would expect convection to contribute significantly to the statistics of subcritical Richardson numbers.

Response: We totally agree with your assessment. $N^2<0$ was not totally filtered out in our analysis, largely due to the fact that a moving average with a bin of 200-m cannot totally avoid the instantaneous convection.

In the revised version, we have separately estimated the occurrence frequency of $0<Ri<Rit$ and $Ri<Rit$ using radiosonde and ERA5, as illustrated in Figure 8. The percentage of $Ri<0$ relative to $Ri<Rit$ is generally less than 20% in the middle and upper troposphere but it is as high as around 40% in the boundary layer.

In addition, in the supporting information, we have added a new figure as Figure S5 to illustrate the occurrence frequency of $0<Ri<Rit,$ which exhibit a similar spatial variability as that of $Ri<Rit$.

[Figure]

**Figure 8**. The percentage of *OF(Ri<0)* relative to *OF(Ri<Rit)* in HVRRS (red) and ERA5 reanalysis (blue).

[Figure]

**Figure S5.** The spatial distribution of the mean *OF(0<Ri<Rit)* in ERA5 reanalysis at 0–2 km a.s.l. (a) and 10–15 km a.s.l. (b). Note that *Rit* is set to 1.

- ○ **HVVRS:**
  - ■ In Fig. 15, the secondary occurrence frequency maximum OF(Ri<1/4) below vertical wind shears of 10 m/s/km could be indicative of such convective contributions. But this is hard to say for sure because all values are vertical averages.

Response: We agree with your assessment. Therefore, we also recalculated the joint distribution of HVRRS-derived wind speed, wind shear, and *OF(0<Ri<Rit)*, and it has

been shown in the supporting information as Figure S8. Compared to Figure 15 in the main text, Figure S8 exhibits a very similar distribution as Figure 15.

[Figure]

**Figure S8.** Joint distribution of HVRRS-derived wind speed, wind shear, and $OF(0<Ri<Rit)$, with a bin size of 5 m/s along the x axis and 5 m/s/km along the y axis. Note that all the relationship is based on the mean result of individual profiles at heights of 10–15 km a.s.l.. The number indicates the matched profile number in each grid.

■ At p.11 L.304 you include convection in the PBL in the interpretation

Response: Yes. As stated above, we recalculated the occurrence frequency of $0<Ri<Rit$ to partly exclude the influence from convection. While the spatial similarity between boundary layer height and $OF(0<Ri<Rit)$ draw a similar conclusion as that of boundary layer height and $OF(Ri<Rit)$

○ **ERA5:** Negative static stability in the model is (as I understand it) quickly resolved through the cloud parametrisation, and therefore rare in the model output. Still, regions of low static stability ($\rightarrow$ small Ri) could be indicative of regions prone to convection, particularly for the adjusted threshold of Rit=1.

Response: Thanks for the informative comments. The initial motivation of the present analysis was that we noticed that more and more studies use ERA5 reanalysis to evaluate turbulence indices in recent years. However, the performance of ERA5 in calculating wind shears and Richardson number was not globally evaluated at present. For $0<Ri<1$, it would be quite difficult for us to differentiate the convective instability or dynamical instability when using ERA5 reanalysis. However, the present analysis could provide a possible reference for further related studies (using ERA5 to calculate wind shears and Richardson number, or low $Ri$ climatology).

■ At p.13 L.361 you mention both KHI and convection in the PBL (diurnal cycle?).

Response: In the revised version, we have removed this phrase. More precisely, both $OF(Ri<Rit)$ and $OF(0<Ri<Rit)$ keeps high consistency with that of planetary boundary layer height (PBLH) over oceans, such as the Pacific Ocean and the Atlantic Ocean. The continental $OF(Ri<Rit)$ in the PBL region could be considerably affected by the vertical resolution of ERA5 reanalysis, which sharply decrease from around 20 m at 0 km a.s.l. to around 120-m at around 2 km a.s.l.

Therefore, two related sentences have been added in the revised version:
"Interestingly, the spatial variation in $OF(Ri<Rit)$ ensembled by years 2017 to 2022 keeps high consistency with that of planetary boundary layer height (PBLH) over oceans, such as the Pacific Ocean near Japan and the Atlantic Ocean near U.S., as shown in Figure S4"
"However, the vertical resolution of ERA5 in the PBL decreases sharply, leading to the fact that the resolution of the PBL data over the region with high elevations can be significantly lower than that of regions with low elevations, which could bring great challenges to the analysis of the impact of topography on low-level $OF(Ri<Rit)$."

- At p.15 L.418 you write the OF(KHI) maximum is related to convective heating.

Response: According to Figure S5b in the supporting information of the updated version, $OF(0<Ri<Rit)$ also exhibits an obvious enhancement over the Niño 3 region. Based on the clues shown in Figure S5b, Figure 9b, and Figure 2c, we argue that the $OF(Ri<Rit)$ anomaly could likely be attributed to the ENSO-related tropical convective heating in the upper troposphere, leading to a low Brunt-Väisälä frequency.

I realize that Fritts et al. (2014) (who you cite in the introduction) discuss how the distinction between dynamic and convective instability can be "challenging an misleading", but I am not sure how this discussion for Lidar measurements in the MLT region is applicable for the troposphere.
Response: Thank. In the updated file, we have specified the conclusion of Fritts et al. (2014) to the MLT region.

p.15 L.422 Would you describe convection in the troposphere generally as weak?
I would suggest to replace OF(KHI) in the manuscript by OF(Ri<Rit), and either expand the discussion and/or leave it to the readers interpretation which process dominates in each region and season. See also the more specific comments further below.
Response: Thanks for the suggestion. We totally agree with your comment. In the updated file, we have replaced all OF(KHI)s with $OF(Ri<Rit)$s throughout all texts and all figures/tables.

**02)**

I am not sure if the interpretation of the data based on the derived correlation coefficients is always justified:

1. The paragraph beginning at p.14 L.399: While I understand the idea behind the analysis and the approach, I am not sure that the interpretation is strongly supported by the data. From the plot it is not really possible to connect the scattered data with the derived linear correlation in Fig. 11. I would assume that a 2d-histogram would reveal that the highest data density is clustered at low SDOR, and that the moderate-to-large SDOR values are negligible for the estimation of the correlation coefficient. Therefore, I would say the interpretation "over mountainous areas a high low-level OF(Ri<1) would be expected" is too strong of a statement based on the data. But of course this is open for discussion.

Response: As suggested, Figure 12 in the updated manuscript (Figure 11 in the old file) has been modified as density histograms. Indeed, the highest data density is clustered at low SDORs. Statistically, the low-level $OF(Ri<Rit)$ is significantly correlated with $OF(Ri<Rit)$ based on hundreds of thousands samples. Complex terrain could locally enhance $OF(Ri<Rit)$ by increasing wind shear or mountain waves. Therefore, in the revised version, we stated that complex terrain may locally enhance $OF(Ri<Rit)$.

[Figure]

**Figure 12**. The association of HVRRS-determined $OF(Ri<Rit)$ with different standard deviations of orography (dimensionless). (a), (b), and (c) are for height ranges of 1–2 km, 3–4 km, and 5–6 km a.g.l., respectively. The correlation coefficients between

*OF*(*Ri*<*Rit*) and standard derivation of orography are marked in the top right corner, where the star superscripts indicate that values are statistically significant ($p$<0.05).

> Furthermore, could you point out which regions in Fig. 8 you mean with the first sentence at p.14 L.399?

Response: According to Fig.9a (Fig.8a in the old file), it is notable that low-level continental *OF*(*Ri*<*Rit*) is dependent on underlying terrains. However, the vertical resolution of ERA5 in the PBL decreases sharply, leading to the fact that the resolution of the PBL data over the region with high elevations can be significantly lower than that of regions with low elevations. It would be inappropriate to use ERA5 to investigate the possible influence of topographies on low-level *OF*(*Ri*<*Rit*). Therefore, we used radiosonde to investigate the association between low-level *OF*(*Ri*<*Rit*) and underlying terrains.

The above concern has been incorporated in the updated text.

> 2. The paragraph beginning at p.15 L.425: Again, I understand the approach of the analysis and the idea behind it, but I would not agree that the conclusion is justified. The correlation is very weak. It is likely dominated by the point cloud at highest data density at low nearsurface-winds. I would not agree that this supports the interpretation at the end of the paragraph.

Response: Thanks for the suggestive comment. We agree with you and totally remove this part in the updated file.

> 3. For comparison: At p.17. L.473 you write: "[The ERA5 determined occurrence frequency of Ri<1/4] is poorly correlated with HVRRS-determined ones at all heights and over all climate zones."
> If this refers to Fig. 7, it might be too strong of a statement considering the overall interpretation of correlation coefficients in the manuscript.

Response: Thanks. This phrase has been modified to be:
"…In addition, it is weak correlated with HVRRS-determined ones at most heights and over most climate zones…"

> **Other scientific comments:**
> **ABSTRACT**
> p.2 L.32 I would prefer to write "KHI is indicated by the critical value of dimensionless Richardson (Ri) number, which is predicted to be ¼ from linear stability analysis." or something along these lines.
> Otherwise a reader without background knowledge might perceive the critical Richardson number as a purely empirical threshold.

Response: Thanks. The suggestion has been incorporated in the Abstract session.

p.2 L34 and L.36 I would suggest to delete the brackets with "137 level" and "10 m", since these values are not directly comparable.
Response: Amended as suggested.

p.2 L.45 In the context of major comment 01, I would suggest to replace "The occurrence frequency of KHI..." with "The occurrence frequency of subcritical Richardson numbers.."
Response: Done as suggested. The replacement has been completed throughout all texts.

p.2 L45 From Fig. 9d and 10k I would argue that there is also a seasonality in the tropics. I see your point that it might overall be less pronounced, but I still would prefer to not exclude the tropics specifically. There are large scale flow features like the Summer Asian Monsoon and the tropical easterly jet or the variability of the Walker circulation (also in relation to ENSO) which could be associated with the tropical seasonality in subcritical Richardson numbers (Roja Raman et al. (2009), Sunilkumar et al. (2015), Kaluza et al. (2021)).
Response: Thanks for the informative suggestion. In the revised file, we stated that seasonal cycles can be detected over all climate zones throughout the text.
   The suggestion about the tropical seasonality has been included in Section 3.3, many thanks!

**INTRODUCTION**

p.3 L.70 Baumgarten and Fritts (2014) focus on the MLT region, and although the statement (or parts of it) might be applicable to the region analyzed here, maybe there is a more general source in literature which describes the environment in which KHI arises.
Response: Based on the suggestion, the statement has been rephrased as:
"…KHI arises preferentially from micro- and mesoscale wind shear intensification, with maximal occurrence frequency near synoptic scale upper-level frontal zones near jet streams, with mountain waves, and above the tops of severe thunderstorms (North et al., 2014)..."

p.3 L.73 Just to clarify, with "the associated large gradient in jet stream" you mean the wind gradients? Or stability gradients? Maybe you could rephrase it to make it more clear.
Response: The gradient has been referred to wind gradients. Moreover, the statement has been modified as:

"…Large wind shear is commonly associated with regions where stability changes rapidly (e.g., near the top of the boundary layer) and the large wind gradient in jet stream (Grasmick and Geerts, 2020)…"

p.3 L.74 "which may increase clear-air turbulence (Williams and Joshi, 2013)."
Do you mean it could increase in a changing climate? If this is the case, maybe rephrase it in an individual sentence.
Response: The climate change could considerably alter wind shears over some regions. Therefore, the statement has been rephrased as:
"…In a changing climate, wind shear in the North Atlantic upper-level jet stream could be increased (Lee et al., 2019), which may increase clear-air turbulence at cruise altitudes…"

p.3 L.81 Maybe remove the sentence "Among others, KHI is one of the most common causes of turbulence throughout the atmosphere from Earth's surface to the lower thermosphere (Fritts et al., 2011; Sharman et al., 2012)." This is somewhat repeated from the sentence before.
Response: Amended.

p.4 L.89 "Values of Ri close to zero favor strong instability, deep billows, and relatively intense turbulence, whereas values of Ri closer to ¼ favor weak instability, shallow billows (Fritts et al., 2011)." I am not sure if this information is necessary for this study.
Response: In the updated file, we also discussed the percentage of $Ri<0$ relative to $Ri<Rit$, as shown in Figure 8. Therefore, we believe this information could be useful to approximate the possible variation of strong or weak turbulence in altitude.

p.4 L.91 "The threshold value of Ri can be potentially used an indicator of turbulence (for instance, Jaeger et al., 2007)."
Suggestion for clarification: "The Richardson number criterion can be applied as a turbulence diagnostic in numerical model output (e.g. Sharman and Pearson, 2017), and it has been used as such in climatological studies on the occurrence of clear air turbulence (Jaeger and Sprenger, 2007)."
Since you cite Jaeger et al. (2007), a more direct comparison with your results in the later sections could be interesting, since Jaeger et al. also presents a Richardson number climatology with an adjusted Rit threshold. And it would put your results in context with the literature.
Kunkel et al. (2019) include a brief discussion on the capability of ECMWF models to resolve subcritical Richardson numbers, and argue as well that Rit=1 might be a good proxy for observed KHI (although only in a case study).

A very recent study on this subject is by Lee at al. (2023), in their climatology on UTLS turbulence diagnostics. They also set the Richardson number threshold from 0-1.

Response: Thanks for the exhaustive suggestion. The above comments have been incorporated into the new file. The related part of text has been modified as:

"…The Richardson number criterion can be applied as a turbulence diagnostic in numerical model output (e.g. Sharman and Pearson, 2017), and it has been used as such in climatological studies on the occurrence of clear air turbulence (Jaeger and Sprenger, 2007). Kunkel et al. (2019) includes a brief discussion on the capability of ECMWF models based on case studies to resolve subcritical Richardson numbers, and argues that the threshold value of $Ri$ ($Rit$) taken as 1 might be a good proxy for observed KHI. A very recent study by Lee at al. (2023) also sets $Rit$ from 0–1 in their climatology on the upper troposphere and lower stratosphere turbulence diagnostics…"

p.4 L.101 "While in numerical models, turbulent dissipation rate, turbulent diffusivity and other parameters representing turbulent mixing efficiency are the most basic parameters, which need to be accurately parameterized to evaluate the impact of turbulent effect on matter and energy distribution (Gavrilov et al., 2005). For this reason, the indices of turbulence, such as large wind shear, small Ri, the negative squared Brunt-väisälä frequency, could be a great tool to characterize turbulence (Jaeger et al., 2007)."

I am not sure how the two sentences and the two citations fit together. Can you maybe rephrase the key statement you want to make here?

Response: This paragraph has been rephrased to be:

"…In numerical models, turbulent dissipation rate, turbulent diffusivity and other parameters representing turbulent mixing efficiency are the most basic parameters, which need to be accurately parameterized to evaluate the impact of turbulence effect on matter and energy distribution (Gavrilov et al., 2005). However, due to the intermittent nature of turbulence it is generally not resolved in (global) numerical weather prediction models, even at nowadays common/states of the art horizontal resolutions of the order of tens of kilometers (Sandu et al., 2019), and it presents a challenge both in observation and numerical modeling (Sharman et al., 2012; Homeyer et al., 2014; Plougonven and Zhang, 2014). For this reason, the indices of turbulence, such as large wind shear, small $Ri$ and the negative squared Brunt-väisälä frequency, could be a great tool to characterize turbulence (Jaeger et al., 2007)..."

p.5 L.121 Is AEOLUS data assimilated in ERA5? At all, or consistently for the time period analyzed?

Response: According to Banyard et al (2021), Aeolus observations have not been assimilated into ERA5. Thus, we removed this statement in the revised version.

Reference:

Banyard, T. P., Wright, C. J., Hindley, N. P., Halloran, G., Krisch, I., Kaifler, B., & Hoffmann, L. (2021). Atmospheric gravity waves in Aeolus wind lidar observations. Geophysical Research Letters, 48, e2021GL092756. https://doi.org/10.1029/2021GL092756

p.5 L.134 Suggestion: "Thus, our objectives are to: (1) Evaluate the performance of ERA5 at different heights and climate zones in estimating wind shear and small Richardson number occurrence frequencies, in comparison with a large high-resolution radiosonde dataset spanning the years from 2017 to 2022."

Response: Done as suggested, thanks.

**1.1 HIGH-RESOLUTION RADIOSONDE DATA SET**

p.6 L.155 Are 115 million radiosonde profiles analyzed in this study? I was wondering because taking the upper limit of the color scale in Fig. S1 times the number of radiosonde stations 3000*434 equals about 1.3 million soundings. Maybe I am missing something.

Response: Very sorry we make a serious mistake. We totally misunderstood the word "million" in the context of English for a long time (We thought it means 10,000). This analysis went through several versions. At the beginning of the analysis, about 1.15 million radiosonde was adopted during years 2016-2022. And then, we removed all ERA5 data on 2016 due to the filled storage of our device (Since ERA5 137 model level data needs huge storage and computation resources). We rechecked our radiosonde dataset during years 2017-2022, and found that about 0.95 million radiosondes have been adopted.

The following Linux terminal displays the total count of radiosonde profiles during years 2017-2022.

```
jian@LAPTOP-2ECIDBLP:/mnt/d/RS_all_10m$ find . -name "*-201[7-9]*" | wc -l
449531
jian@LAPTOP-2ECIDBLP:/mnt/d/RS_all_10m$ find . -name "*-202*" | wc -l
495860
```

**1.2 ERA5 REANALYSIS AND THE COLLOCATION PROCEDURE**

p.7 L.177 I would suggest to delete the part "...and 500 m in the lower stratosphere." It is not really necessary nor supported by Fig. S2.

Response: Amended.

p.7 L.184 If the orographic gravity wave dispersion is an ECMWF product, it should be listed here as well.

Response: Amended. The paragraph has been modified as:

"…In addition, the standard deviations of orography (SDOR) and the gravity wave dissipation due to the effects of stress associated with unresolved valleys, hills and mountains in ERA5 reanalysis are extracted…"

**2.3 THE OCCURRENCE FREQUENCY OF KHI AND ITS UNCERTAINTY**

p.7 L.190 I am missing an introduction of the critical Richardson number as a quantity which can be derived from linear theory. I personally would include this information in a study which focuses on the Richardson number.

Response: The related lines have been modified as:

"Based on a linear theory, the threshold $Ri$ ($Ri_t$) defines the threshold where the air flow changes from stability to turbulence, and it is usually suggested to be 1/4 (Haack et al., 2014). $Ri$ is formulated as…"

p.7 L.193 please give the definition of the Brunt-Väisälä frequency and the vertical wind shear.

Response: Amended.

p.7 L.198 "The occurrence frequency of KHI is defined as the ratio of Ri<1/4 relative to all Ri calculations at a specified time period or height interval."
I don't think this sentence is necessary, and it can be misleading since you vary the threshold for the Richardson number.

Response: This statement has been deleted.

p.7 L.200 "In Eq.(1), the length scale of overbar could potentially impact the value of Ri, and eventually, the occurrence frequency of KHI. In addition, the critical value of Ri and the vertical resolution of archived radiosonde could also cause the change in Ri values."
I would suggest to rephrase these two sentences to make the key point more clear.
Suggestion: "The Richardson number calculated from Eq.(1) depends on the vertical resolution of the underlying data, as well as on the averaging interval. Ultimately, this influences the estimated occurrence frequency for subcritical Richardson numbers as a proxy for KHI. We resample the HVRRS data..."

Response: Very thanks for the patient modification! The correction has been made as suggested.

**2.4 GRAVITY WAVE ENERGY**

p.8 L.226 "Only the perturbations with vertical wavelengths of 0.3–6.9 km are considered as GWs (Wang and Geller, 2003)."
For clarification, is this due to the vertical limitation from 2-8.9 km (i.e. 6.9 km as an

upper limit) and the effective vertical resolution of 150m (i.e., half a wavelength) in Wang and Geller (2003)? Or do I misread this information?
How does this apply to your data?
Maybe this information should be provided after the description of the "tropospheric segment".

Response: The bandpass of filter was not well determined in different literature. For instance, Chen et al. (2019, CD) applied a passband of 1–4.5 km to the disturbance field to extract wave energy. In our recent study on tropospheric gravity waves carried out in 2020/2021 (Zhang et al., 2022, JGR-Atmospheres), we tested several bandpass filters and found that these filters have little effect on the overall distribution of gravity wave energy.

Prof. Wang and Prof. Geller did a lot of pioneering work on radiosonde-determined GWs. They referred the wavelength to be 0.3–6.9 km on their several studies, for instance, Wang and Geller (2005, JAS). Most GW energy concentrates on waves with vertical wavelength over 1 km, and the high-pass filter with a length of 0.3 km has limited impact on GW energy. In addition, as stated by Alexander (1998), the largest wavelength is not well determined, which is acknowledged as the radiosonde's "observational filter". This concern has been included in the main text as:

"…The mean vertical wavelength of GWs is about 2 km (Wang et al., 2005), and therefore, the lowermost threshold of 0.3 km could have little influence on the GW energy. However, the retrieval of the largest wavelength is not well determined, which is acknowledged as the radiosonde's "observational filter" (Alexander, 1998)…"

**3   RESULTS AND DISCUSSION**

p.9 L.250 please provide a link to Table 1 if the value 4m/s/km is taken from this table. If it is not the case, please indicate how the value was derived.
Response: As compared to the HVRRS, these shears are slightly underestimated by 5.37 m/s/km, based on all sounding measurements (Fig.2b). The correction has been made in the updated file.

p.9 L.251 "However, the oceanic shear is hard to be quantitatively assessed by a large number of in-situ radiosonde stations, with this aspect likely being evaluated by the ship-based radiosonde." I am not sure if this sentence is necessary, from the data description it is clear that radiosonde soundings over the oceans are sparse.
Response: This statement has been delated in the revised version.

p.9 L.253 "Over the tropical oceans, Savazzi et al. (2022) found the wind bias between EUREC4A field campaign and the ERA5 reanalysis varies greatly from day to day, attributing to the bias in wind forecasting in the ERA5 reanalysis."

Again just a suggestion, but I personally would remove this sentence as well, since Savazzi et al. (2022) has been cited in the introduction, and the statement made here is not directly linked to the analysis. It might increase the reading flow if the manuscript is a bit more compact.
Response: This phrase has been removed as suggested.

p.10 L.258 Maybe provide a link to Fig. 2b at the end of the sentence.
Response: Amended.

p.10 L.259 Suggestion: "It is noteworthy that shear in the ERA5 reanalysis at heights of 10–15 km a.s.l. is significantly underestimated compared to the HVRRS, especially at middle latitudes, with a mean absolute error for all stations of about 8 m/s/km (Table 1).
(Assuming the value 8m/s/km is taken from Table 1)
Response: Thank. Point taken.

p.10 L.265 Suggestion: "Following Houchi et al. (2010), the monthly shears over seven typical climate zones are separately investigated (Fig. 3), which are defined as follows: polar (70°–90°), mid latitudes (40°–70°), subtropics (20°–40°), and tropics (20°S–20°N).Over the polar region HVRRS-based shears are exceptionally strong in the stratosphere (Fig.3a, g), which could be attributed to the stratospheric polar jet."
I think it is clear from the data description and the plots that both Hemispheres are analyzed separately.
Response: Thanks for the modification! Point taken.

p.10. L.274 Is the value 16 m/s/km taken from Table 1? If this is the case, please provide a link in the text. If it is not the case, please indicate how the value was derived.
Response: The link of Table 1 has been added.

p.10. L.275 Maybe remove "in the Northern/Southern Hemisphere", it is again implied.
Response: Point taken.

p.10 L.277 and Fig.3 h-n: Could you please comment on the vertically aligned wind shear structures in the ERA5 derived wind shear in summer 2017 and beginning of 2020? Are these real or artefacts due to the data processing? They look suspicious particularly since the figure shows monthly averages.
Furthermore, why do the regions of missing values in Fig. 3g and 3n differ (mid 2017 and end 2022)?
Response: This was caused by data processing errors. Two of files on 2017 and 2020

were only partly downloaded. The following CDO operation raised errors when calculating the monthly mean wind shears. We have fixed these errors and updated the figure.

[Figure]

**Figure 3.** Monthly mean wind shears during years 2017–2022 in HVRRS (a–g) and ERA5 reanalysis (h–n) at different climate zones. The ERA5 derived wind shears are spatially and temporally collocated with those of HVRRS. NH=Northern Hemisphere; SH=Southern Hemisphere.

p. 10 L.279: "…which is about 3 km lower than that in the HVRRS."
Just out of interest, would you say this is the same shear layer shifted downward, i.e., a 3 km altitude bias in the dynamic structure of the UTLS?
Or are the shear magnitudes at the altitude of the maximum in ERA5 comparable to the ones in the HVRRS and the model fails to resolve the further increasing shear in the stratosphere?
Response: According to the fact that the vertical resolution of ERA5 in the UTLS smoothly decreases with height, we trend to suppose that model might be failed to resolve the further increasing shear in the stratosphere. Therefore, we have added a statement in the revised version as:
"…One possible reason might be that the model fails to resolve the further increasing shear in the lower stratosphere…"

p.10 L.283 I would suggest to remove the second part of the sentence, and just write: "The comparison between HVRRS-based and ERA5-based shears at three typical regimes are tabulated in Table 1."

Because the altitude range from 10-15 km is not the middle and upper troposphere, particularly at high latitudes.
Furthermore, if the mean absolute errors at page 9 and 10 (4m/s/km, 8m/s/km and 16m/s/km) refer to Table 1, then it would be better to introduce Table 1 beforehand.
Response: Thanks! Amended as suggested.

p.11 L.292 (Fig. S3): Just a general thought, it could be interesting to see this comparison for the troposphere and the stratosphere separately, or for different altitude intervals.
Response: Point taken. The comparison has been conducted at four height intervals, they are, 0–5 km a.s.l., 5–10 km a.s.l., 10–15 km a.s.l., and 15–20 km a.s.l., as displayed in the revised Figure S3.

[Figure]

**Figure S3.** The joint distributions of HVRRS-retrieved and ERA5-determined squared Brunt-väisälä frequency together with the linear regression (red line) at heights of 0–5 km a.s.l. (a), 5–10 km a.s.l. (b), 10–15 km a.s.l. (c), and 15–20 km a.s.l. (d). The light red shadow denotes a significance of 95%. The Brunt-väisälä frequency is averaged during the whole study period. The ERA5 derived $N^2$ is spatially and temporally collocated with that of HVRRS.

p.11 L.304 "The high occurrence frequency in the PBL regime could be likely related to the convective activity that leads to a negative $N^2$."
Please see major comment 01.
Response: Based on the spatial distribution of $OF(Ri<Rit)$ (Figure 9a), $OF(0<Ri<Rit)$

(Figure S5a), and wind shear (Figure 2b) at 0–2 km a.s.l., the high occurrence frequency of subcritical $Ri$ could be attributed to negative or small $N^2$. The clue in Figure S3a could also support this conclusion. In the updated file, we stated that

"…The high occurrence frequency in the PBL regime could be likely related to the negative or small $N^2$…"

p.11 L.307 "In addition, an obvious seasonal cycle of occurrence frequencies is revealed by HVRRS in the middle and upper troposphere and has a maximal in the spring season (March–April–May), which is consistent with the finding in Zhang et al. (2019).

The maximum in the middle troposphere is during spring, the upper tropospheric maximum is also evident in winter I would say.

Response: The modification has been made as suggested.

p.11 L.313 What does the value 8% refer to? Is it an estimation from Fig.4, or was it calculated from the data? If the latter is the case, which data exactly?

Response: In the updated file, we rephrased this sentence to be:

"…However, the ERA5 reanalysis does not provide such a seasonal cycle pattern, and the occurrence frequency of $Ri$<1/4 is significantly underestimated by around 8% (Fig.4b)…"

p.12 L.319 In Fig. 5 it says "10-15 km".

Response: The correction in the main text has been made.

Figure 6: Maybe line plots would increase the readability compared to scatter plots. Or smaller markers.

Response: Point taken. Figure 6 has been replaced with line plots.

[Figure]

**Figure 6.** The altitude variation of the occurrence frequency of *Ri* below certain thresholds (0.25, 0.5, 1, 1.5, and 2) in ERA5 reanalysis in various climate zones. The ERA5 derived *Ri* is spatially and temporally collocated with that of HVRRS. The occurrences of *Ri*<1/4 in HVRRS are overlapped with red lines.

p.13 L.345 Please see major comment 01. Of course this is open to discussion, but I would suggest to replace OF(KHI) in the following analyses with $OF(Ri_{ERA5}<1)$ or something similar. I don't think this lessens the impact of the results.
Response: Point taken. All *OF*(KHI)s have been replaced throughout all texts, tables, and figures.

p.13 L.358 I am not sure if I can follow this argumentation. Could you point out exemplary regions for the spatial consistency?
How meaningful is it to compare multi-year averages for these two highly variable quantities?
Response: The boundary layer height in ERA5 reanalysis is based on the bulk Richardson number algorithm, which is closely associated with the *OF*(*Ri*<*Rit*). Over ocean, the spatial variation of *OF*(*Ri*<*Rit*) and PBLH is quite similar. However, the continental *OF*(*Ri*<*Rit*) at 0–2 km a.s.l. is significantly influenced by the underlying terrain, especially over region with high elevations.

p.13 L.361 See major comment 01.
Response: The statement has been deleted due to the change of context.

p.13 L.366 Suggestion: "In the free troposphere the spatial-temporal variability of OF(RiERA5<1) keeps high consistency with OF($Ri_{HVRRS}$<1/4) over all climate zones." I would refrain from specifying the altitude range up to 30km because of the stratospheric polar maxima in the HVRRS.

Response: Point taken, thanks.

Figure 9:
- What are the vertically aligned structures in the ERA5 analysis at the beginning of 2020?
- Why is the ERA5 data here (and in Fig. 4) cropped at 29 km?
- Why are there missing data patches in the beginning of 2022 in the ERA5 data across all climate zones?
- Why are contour lines included in the ERA5 plots but not in the HVRRS? Maybe keep them similar for better comparison.

Response: The incomplete ERA5 files has been replaced. Also, Figure 10 has been replotted. While the contour lines in (a-g) are too dense to be read, as shown in Figure A. Therefore, the contour lines in Figure 10a-g are still missing.

[Figure]

**Figure A.** The monthly averaged $OF(Ri<Rit)$ in the HVRRS (a–g).

[Figure]

**Figure 10**. The monthly averaged *OF(Ri<Rit)* in the HVRRS (a–g) and ERA5 reanalysis (h–n) in seven climate zones. NH=Northern Hemisphere; SH=Southern Hemisphere.

p.14 L.372 I would suggest to remove the sentence "For regions without high-resolved wind and temperature measurements, the ERA5 model product could be a good choice to represent the thermodynamic instability of background atmosphere."
This has been introduced in the motivation for the study.
Response: Point taken.

p.14 L.377 Suggestion: "The seasonal variation of OF(Ri<Rit) with Rit,HVRRS=1/4 and Rit,ERA5=1 for all climate zones is further analyzed in Figure 10. In the mid latitudes and subtropics, the OF(Ri<Rit) exhibits maximum values in the PBL, as well as a local minimum in the middletroposphere and a local maximum at altitudes around 9 km. In the stratosphere the occurrence frequencies decrease to values of the order of 1% (Fig.10b,c,e,f)."
There is a significant variability in the vertical profiles across the different data sets, seasons and climate zones, and the single-value average occurrence frequencies in the current version of this paragraph might be confusing.
Response: Thanks for the correction. Modification has been made.

Figure 10: I would suggest to remove the gray shaded areas, I don't think they are necessary for the analysis.
Please adjust the x-axis range in Fig. 10k and 10l so that the JJA maxima are not cut off.

Response: Amended as suggested.

Response: Amended as suggested

Response: As stated above, the modification concerning underly terrains has been made.

Figure 11: Could you plot the data as a binned 2d-histogram? What is the unit at the x-axis?
Response: The Figure has replotted as 2d-histrogram. SDOR is dimensionless, which is indicated in the caption of Figure 12.

Response: Thanks. We recalculated all ERA5 data to estimate $OF(0<Ri<Rit)$ (Figure S5 in the updated supporting information) and found that $OF(0<Ri<Rit)$ also has a significant enhancement over the Niño 3 region.

Response: The paragraph has been modified to be:
"In the free troposphere where the convection activity is generally weak, KHI is preferentially generated from strong wind shear, which may be closely associated with mean flows and wave activities."

Response: We agree with your comment. This part has been totally deleted in the updated file.

p.16 L.436 "Overall, large OF(KHI) always corresponds to strong GW activities and large wind shears, likely indicating that GW activity is crucial for the occurrence of KHI."

Is this derived from Fig. 14b? If so please link the Figure in the text.

I also think that a more detailed description would be helpful. What do you define as "strong GW activity" in the plot? Where do you see the correlation between strong GW activity and OF(Ri<Rit)? Maybe I am misreading the plot, but I find it hard to follow the conclusion.

Response: Based on the comment, we modified the statement to be"

"…The joint distribution of $OF(Ri<Rit)$ with GW energy and wind shear indicates that large $OF(Ri<Rit)$ (for instance, larger than 10%) generally corresponds to GW energy larger than 10 J/kg or wind shear exceeds 14 m/s/km (Fig. 14b)..."

p.16 L.440 Is the orographic gravity wave dissipation an ECMWF product? If this is the case please include it in the data description.

And just for clarification, is this the quantity derived from the parametrised gravity wave drag due to subgrid-scale orography? So you identify regions and time steps (months) of strong resolved gravity wave activity based on the parametrised non-resolved gravity waves?

Response: In the website of ECMWF, it says "It (GW dissipation) is calculated by the ECMWF Integrated Forecasting System's sub-grid orography scheme, which represents stress due to unresolved valleys, hills and mountains with horizontal scales between 5 km and the model grid-scale". In the data description, we have added the related description about GW dissipation.

This parameter cannot represent resolved gravity waves, which could be still a challenge for general circulation models. We extracted this parameter was to investigate the possible association between mountain barrier and $OF(Ri<Rit)$ by the propagation or dissipation of mountain waves, since radiosondes can only provide limited information on global mountain waves.

Figure S5: According to the text (p.16 L.443) the correlation is based on monthly averaged values (of the gravity wave dissipation and the OF(Ri<Rit)). Please include this information in the description of Fig. S5.

Response: Amended.

Again just for clarification: What years are the data basis for this plot? 2017-2022? So about 60 data points are correlated at each grid point?

Response: Yes, these data cover time period of 201701 to 202210, and 70 data points were used. This information has been included in the caption of Figure S7 in the

updated files

So during months with strong parametrised gravity wave activity, a strong activity of resolved gravity waves can be expected, which then modify the flow and stability parameters of the resolved flow, and result in an enhanced occurrence frequency for low Richardson numbers.
However, in regions where according to Fig. 8b small Richardson numbers are rare (e.g. over the Rocky Mountains, the Andes, Scandinavia, the Alps).
It would be interesting to see a time series of the monthly averaged gravity wave dissipation and the OF(Ri<Rit), for example over the Rocky Mountains. To get an impression of what this correlation means in terms of absolute values and the variability of OF(Ri<Rit). But this is maybe beyond the scope of this study, since it is already pretty comprehensive.

Response: As suggested, we added four sub-plots around Figure S7a, to demonstrate the temporal variation of GW dissipation and $OF(Ri<Rit)$. Two of them locate at the Rocky Mountain and the rest of them locate at the Andes Mountain. These four grids were randomly selected. Based on these sub-plots, the monthly averaged parametrized GW dissipation keeps high consistency with the monthly $OF(Ri<Rit)$.

[Figure]

**Figure S7**. The correlation coefficient between monthly averaged ERA5-based orographic GW dissipation and monthly ERA5-based $OF(Ri<Rit)$ at pressure levels of

375 hPa (a) and 205 hPa (b) during time period of January 2017 to October 2022. where plus signs indicate that the values are statistically significant (*p*<0.05). Four subplots around (a) display the monthly variation of orographic GW dissipation (red) and *OF(Ri<Rit)* (black) over two grids of the Rocky Mountain and two grids of the Andes Mountain. The coefficient at continuous heights from the ground up to 30 km is further displayed in (c), where the light red shadow denotes a significance of 95%.

Figure 15: I am not sure if I read this plot correctly. The bin size for the filled color contour (i.e., the OF(RiHVRRS<1/4)) is apparently something of the order of 1 m/s in the x-axis, and 1 m/s/km in the yaxis. If this is the case, shouldn't most of the filled contour display missing values, within 5x5 bins where only 1 or 2 matched profiles are located?

Response: We checked the plot codes of Figure15 as well as Figure 14b and found that one of line was badly designed, which can raise index error when the grid has no matched data. We redesigned the code and replotted Figure15 and Figure 14b.

[Figure]

**Figure 14**. Geographical distribution of mean tropospheric GW total energy obtained from the HVRRS (a). The latitudinal variation of mean energy in a grid cell of 5° latitude (b). The joint distribution of *OF(Ri<Rit)* with GW energy and wind shear (c). The *OF(Ri<Rit)* and wind shear are derived from individual HVRRS profiles and vertically averaged over the tropospheric segment that is used for GW study. The numerical number in (c) indicates the matched profile number in each grid, using a bin size of 2 J/kg along the x axis and 2 m/s/km along the y axis.

[Figure]

**Figure 15.** Joint distribution of HVRRS-derived wind speed, wind shear, and *OF(Ri<Rit)*, with a bin size of 5 m/s along the x axis and 5 m/s/km along the y axis. Note that all the relationship is based on the mean result of individual profiles at heights of 10–15 km a.s.l.. The number indicates the matched profile number in each grid.

p.17 L.461 "The occurrence of KHI is potential crucial for many implications, such as aircraft, mass transfer, and climate change, just name a few, but it is very hard to be globally understood due to its fine structure."

Please rephrase this sentence.

Aircraft → aircraft safety?

Climate change → how?

Response: The sentence has been rephrased to be:

"…The occurrence of KHI is potential crucial for many implications, such as aircraft safety and mass transfer, but it is very hard to be globally understood due to its fine structure..."

p.17 L.463 Suggestion: "This study uses the ERA5 as the latest reanalysis product from the ECMWF as well as a comprehensive data set of HVRRS radiosonde soundings to globally characterize the distribution of low Richardson numbers as a proxy for the occurrence of KHI, for the years 2017 to 2022."

Response: Thanks! Amended.

p.17 L.471 I would suggest to remove the sentence "The underestimation therefore influences the performance of KHI analysis."

Response: Amended as suggested

p.17 L.473 See major comment 02.

Response: The statement has been modified as:

"…In addition, it is weak correlated with HVRRS-determined ones at most heights and over most climate zones…"

p.17 L.479 "...especially in the middle and upper troposphere over midlatitude and subtropic regions in the Northern/Southern Hemisphere."
Why especially in these regions?
Response: Based on Table 4, $OF(Ri<Rit)$s at 10–15 km a.s.l. over midlatitude and subtropic regions revealed by ERA5 and radiosonde are quite similar in terms of magnitude.

p.17 L.484 to p.18 L.490
This paragraph might need to be reworked based on how the manuscript was changed after the review.
Response: The correlation between $OF(Ri<Rit)$ and near-surface wind has removed from manuscript, as stated above. The joint distributions among $OF(Ri<Rit)/OF(0<Ri<Rit)$, gravity waves and background wind speed have been reworked. Accordingly, the conclusion has been revised.

p.18 L.491 Maybe this is a personal preference, but I would not write the final paragraph in subjunctive.
For example (again just a suggestion): "Those findings are valuable for pointing out the performance of the ERA5 reanalysis in terms of resolving low Richardson numbers as a proxy for KHI, in comparison with a near-global high-resolution radiosonde measurement." Same for the last sentence.
Response: Thanks. Amended as suggested.

p.19 L.516 Please be more specific and include both data sets (ERA5 and HVRRS).
Response: Done

**Technical corrections (typos etc.):**
Throughout the manuscript: Maybe it should say "a.s.l." instead of "a.s.l"?
Response: Done

p.3 L.67 Remove the sentence "In addition, GW breaking has been identified as important sources of instability", this is already stated one sentence earlier.
Response: Done

p.4 L.108 I would suggest to either write "The Richardson number is estimated by the…" or "Ri is estimated by the..."
Response: Done

Response: Done

p.7 L.179 Suggestion: "Compared to ERA5, the HVRRS does not provide global seamless observations."

Response: Done

p.9 L.248 . "Large wind shear is common in regions where stability changes rapidly (Grasmick and Geerts, 2020)." I would suggest to delete the sentence as it is already in the introduction.

Response: Amended

Response: Done

Response: Done

p.11 L.311 Suggestion: "However, the ERA5 reanalysis does not provide such a seasonal cycle pattern.."

Response: Done

p.12 L.331 "underestimated" instead of "undervalued". Same for "overvalued".

Response: Done

Response: Done

p.13 L.371 I am not sure about the word "backbone". Maybe replace it with "large scale structure" or something along those lines?

Response: Done

p.15 L.408 Replace "associated with El Niño Southern Oscillation (ENSO) events." with "associated with the El Niño Southern Oscillation (ENSO)."

Response: Done

Response: Done

p.17 L.468 "vertical wind shear" instead of "shears"

Response: Amended.

**References:**

Dutton, J. A., & Panofsky, H. A. (1970). Clear air turbulence: A mystery may be unfolding. Science, 167(3920), 937–944. https://doi.org/10.1126/science.167.3920.937

Kaluza, T., Kunkel, D., & Hoor, P. On the occurrence of strong vertical wind shear in the tropopause region: A 10-year ERA5 northern hemispheric study. Weather and Climate Dynamics, 2(3), 631–651. https://doi.org/10.5194/wcd-2-631-2021, 2021.

Lee, J. H., Kim, J.-H., Sharman, R. D., Kim, J., & Son, S.-W. Climatology of Clear-Air Turbulence in upper troposphere and lower stratosphere in the Northern Hemisphere using ERA5 reanalysis data. Journal of Geophysical Research: Atmospheres, 128, e2022JD037679. https://doi.org/10.1029/2022JD037679, 2023.

Roja Raman, M., Jagannadha Rao, V. V., Venkat Ratnam, M., Rajeevan, M., Rao, S. V., Narayana Rao, D., and Prabhakara Rao, N.: Characteristics of the Tropical Easterly Jet: Long-term trends and their features during active and break monsoon phases, J. Geophys. Res.-Atmos., 114, 1–14, https://doi.org/10.1029/2009JD012065, 2009.

Sharman, R. D., & Pearson, J. M. Prediction of energy dissipation rates for aviation turbulence. Part I: Forecasting nonconvective turbulence. Journal of Applied Meteorology and Climatology, 56(2), 317–337. https://doi.org/10.1175/JAMC-D-16-0205.1, 2017.

Sunilkumar, S. V., Muhsin, M., Parameswaran, K., Venkat Ratnam, M., Ramkumar, G., Rajeev, K., Krishna Murthy, B. V., Sambhu Namboodiri, K. V., Subrahmanyam, K. V., Kishore Kumar, K., and Shankar Das, S.: Characteristics of turbulence in the troposphere and lower stratosphere over the Indian Peninsula, J. Atmos. Sol.-Terr. Phys., 133, 36–53, https://doi.org/10.1016/j.jastp.2015.07.015, 2015.

Response: Many thanks for the provided references

---

## Author Comment (AC2)

**Response to Reviewers #2 Comments**

We thank the associate editor, editor and two anonymous reviewers for their thoughtful and exhaustive comments and suggestions, which significantly help us to improve the quality of the manuscript. In this revised manuscript, we have revised the manuscript accordingly. Below, we indicate the original comment of the respective reviewer in blue and our point-to-point response is denoted in black.

Before addressing the comments, we would like to express our sincere gratitude to the reviewers for their exceptionally informative, constructive, and detailed comments.

**Review report on the paper "Occurrence frequency of Kelvin Helmholtz instability assessed by global high-resolution radiosonde and ERA5 reanalysis" by Shao et al. submitted to the journal Amospheric Chemistry and Physics.**

Overview This paper describes the spatial and temporal variability of instabilities in the atmosphere from a huge radiosonde (RS) database (115 million profiles (?) from 434 stations!). The resulting statistics from the radiosonde analysis are compared to those obtained from ERA5 reanalyses. The authors observe a severe underestimation of the vertical shear from ERA5. They find a better agreement between the instability climatology from RS and ERA5 by taking a threshold Ri < 1 when using ERA5. The authors also observe a positive correlation with the standard deviation of the orography. The analysis of the RS and ERA5 data is interesting and the climatological results seem very convincing. The comparison RS/ERA5 is also interesting. For these reasons, this work deserves publication. However, the interpretation of the results does not always seem to me to be correct. For instance, the fact of interpreting all instabilities detected from the Ri < 1/4 criterion as the result of Kelvin-Helmholtz instabilities is not justified, in particular in the boundary layer, or in the tropical troposphere. Therefore, substantial modifications seem to me necessary, if only the title.

I therefore recommend that this article be published with some substantial modifications.

Response: We sincerely appreciate the reviewers for their thorough and insightful comments and assessments, which have greatly benefited our present work as well as our future research. We learned a lot from your suggestive comments. In the following, we address your comments in detail, and we have revised some parts of our work based on your constructive suggestions, aiming to achieve your desired outcomes.

**Major comments**

1) lines 190-217: about the method for estimating the gradients from the HVRRS profiles. If I understand correctly, a moving average of the estimated gradients is performed over height segments of 200 m. But over what vertical scale are the gradients evaluated (which are then averaged)? Are they calculated over 10 m differences? If so, why this choice if one purpose is to compare with the estimates from ERA5? Why not estimate the vertical gradients of HVRRS on vertical scales comparable to the model (100-400 m) for comparison? Moreover, it would be simple to adapt the resolution of the HVRRS gradients to the resolution of the model according to the altitude domains considered.

Figure 1 is illuminating on that purpose in showing that the estimates of the occurrence frequency (OF) of Ri on a 100 m scale, without averaging, is little different from that estimated on 10 m differences averaged over 200 m.

Such a resolution (10 m averaged on 200 m segments) is relevant to establish the climatology of instability occurrences (Ri < 1/4) obtained from HVRRS, but is arguably questionable for comparison to the climatology deduced from ERA5.

Response: Very thanks for the comment. We totally agree with your assessment. In the revised version, we carried out the comparison based on two scenarios: (1) the comparison with 10-m radiosonde at all heights from near-ground up to 30 km; (2) the comparison with radiosonde at four specified height intervals with comparable vertical resolution. We hope these two versions of comparison could leave more spaces for readers.

In the updated files, Tables 2 and 3 illustrate the comparison of vertical wind shears and occurrence rate of low $Ri$ at comparable vertical resolutions. The vertical resolution of radiosonde at 0.8–1.3 km, 2.2–3.2 km, 6–15 km, and 20–21 km a.s.l. was resampled to 100-m, 200-m, 300-m, and 400-m, respectively. The results suggest that even at a comparable vertical resolution, the vertical wind shear and $OF(Ri<1/4)$ in ERA5 reanalysis might be significantly underestimated. At these four height intervals, $0.5<Rit<1.5$ might be a more reasonable choice, rather than 1/4. In general, $Rit=1$ could be a proper choice for ERA5 reanalysis.

Accordingly, the related texts and tables has been added in the updated manuscript.

**Table 2**. The occurrence rate of low $Ri$ at 0.8–1.3 km a.s.l. (a), 2.2–3.2 km a.s.l. (b), 6–15 km a.s.l. (c), and 20–21 km a.s.l. (d). The critical $Ri$ ($Rit$) is 1/4 for radiosonde, and it increases from 1/4 to 2 for ERA5 reanalysis. Note that HVRRS data were vertically resampled to 100-m, 200-m, 300-m, and 400-m at these four height intervals to match with the ERA5 reanalysis. In addition, the moving average number in Eq.(1) is 0. RS stands for radiosonde.

**(a) Frequency of low *Ri* at 0.8–1.3 km a.s.l. (%) / Vertical resolution of RS is 100-m**

| | Polar (NH) | Midlatitude (NH) | Subtropics (NH) | Tropics | Subtropics (SH) | Midlatitude (SH) | Polar (SH) |
|---|---|---|---|---|---|---|---|
| RS, *Rit*=1/4 | 15.20 | 24.25 | 22.86 | 13.92 | 22.09 | 22.43 | 20.77 |
| ERA5, *Rit*=1/4 | 2.55 | 8.88 | 6.37 | 2.19 | 6.80 | 4.47 | 2.94 |
| ERA5, *Rit*=0.5 | 3.77 | 12.06 | 9.63 | 3.65 | 11.91 | 7.95 | 7.22 |
| ERA5, *Rit*=1 | 8.54 | 21.22 | 20.48 | 8.27 | 25.45 | 18.21 | 15.78 |
| ERA5, *Rit*=1.5 | 14.18 | 29.62 | 30.44 | 12.88 | 36.07 | 27.97 | 23.22 |
| ERA5, *Rit*=2 | 19.44 | 36.58 | 38.32 | 17.20 | 43.72 | 36.00 | 29.68 |

**(a) Frequency of low *Ri* at 2.2–3.2 km a.s.l. (%) / Vertical resolution of RS is 200-m**

| | Polar (NH) | Midlatitude (NH) | Subtropics (NH) | Tropics | Subtropics (SH) | Midlatitude (SH) | Polar (SH) |
|---|---|---|---|---|---|---|---|
| RS, *Rit*=1/4 | 3.04 | 6.22 | 9.00 | 5.67 | 9.71 | 4.29 | 3.98 |
| ERA5, *Rit*=1/4 | 0.24 | 0.60 | 1.00 | 1.30 | 2.26 | 0.26 | 0.1 |
| ERA5, *Rit*=0.5 | 0.37 | 1.03 | 1.96 | 2.10 | 4.22 | 0.50 | 0.18 |
| ERA5, *Rit*=1 | 1.16 | 3.26 | 6.35 | 5.20 | 10.00 | 2.20 | 0.91 |
| ERA5, *Rit*=1.5 | 2.77 | 6.75 | 12.20 | 9.00 | 16.31 | 5.60 | 2.68 |
| ERA5, *Rit*=2 | 5.02 | 10.85 | 18.05 | 13.03 | 22.45 | 9.84 | 5.10 |

**(c) Frequency of low *Ri* at 6–15 km a.s.l. (%) / Vertical resolution of RS is 300-m**

| | Polar (NH) | Midlatitude (NH) | Subtropics (NH) | Tropics | Subtropics (SH) | Midlatitude (SH) | Polar (SH) |
|---|---|---|---|---|---|---|---|
| RS, *Rit*=1/4 | 0.76 | 2.24 | 3.91 | 5.98 | 4.46 | 1.98 | 0.59 |
| ERA5, *Rit*=1/4 | 0.10 | 0.38 | 0.54 | 1.46 | 0.56 | 0.24 | 0.05 |
| ERA5, *Rit*=0.5 | 0.32 | 1.16 | 1.95 | 4.36 | 2.10 | 0.93 | 0.15 |
| ERA5, *Rit*=1 | 1.37 | 4.33 | 7.72 | 13.14 | 8.89 | 3.51 | 0.61 |
| ERA5, *Rit*=1.5 | 2.93 | 8.31 | 14.54 | 21.78 | 17.05 | 6.76 | 1.38 |
| ERA5, *Rit*=2 | 4.70 | 12.35 | 20.91 | 29.28 | 24.55 | 10.02 | 2.32 |

**(d) Frequency of low *Ri* at 20–21 km a.s.l. (%) / Vertical resolution of RS is 400-m**

| | Polar (NH) | Midlatitude (NH) | Subtropics (NH) | Tropics | Subtropics (SH) | Midlatitude (SH) | Polar (SH) |
|---|---|---|---|---|---|---|---|
| RS, *Rit*=1/4 | 0.03 | 0.07 | 0.13 | 0.04 | 0.04 | 0.10 | 0.07 |
| ERA5, *Rit*=1/4 | 0.01 | 0.02 | 0.01 | 0.02 | 0.02 | 0.03 | 0.04 |
| ERA5, *Rit*=0.5 | 0.02 | 0.03 | 0.01 | 0.02 | 0.03 | 0.04 | 0.04 |
| ERA5, *Rit*=1 | 0.03 | 0.05 | 0.04 | 0.05 | 0.05 | 0.08 | 0.04 |
| ERA5, *Rit*=1.5 | 0.04 | 0.11 | 0.13 | 0.19 | 0.09 | 0.17 | 0.04 |

| ERA5, $Ri_t$=2 | 0.05 | 0.21 | 0.32 | 0.55 | 0.18 | 0.30 | 0.05 |
| --- | --- | --- | --- | --- | --- | --- | --- |

**Table 3**. Vertical wind shears at 0.8–1.3 km a.s.l. (a), 2.2–3.2 km a.s.l. (b), 6–15 km a.s.l. (c), and 20–21 km a.s.l. (d). Note that HVRRS data was vertically resampled to 100-m, 200-m, 300-m, and 400-m at these four height intervals to match with the ERA5 reanalysis. RS stands for radiosonde.

**(a) Wind shear at 0.8–1.3 km a.s.l. (m/s/km) / Vertical resolution of RS is 100-m**

| | Polar (NH) | Midlatitude (NH) | Subtropics (NH) | Tropics | Subtropics (SH) | Midlatitude (SH) | Polar (SH) |
| --- | --- | --- | --- | --- | --- | --- | --- |
| RS | 12.50 | 13.63 | 11.80 | 9.83 | 13.54 | 13.06 | 13.85 |
| ERA5 | 5.43 | 5.92 | 6.47 | 4.83 | 7.02 | 6.71 | 6.05 |

**(b) Wind shear at 2.2–3.2 km a.s.l. (m/s/km)/ Vertical resolution of RS is 200-m**

| | | | | | | | |
| --- | --- | --- | --- | --- | --- | --- | --- |
| RS | 8.31 | 9.09 | 9.24 | 9.08 | 9.45 | 9.39 | 10.00 |
| ERA5 | 3.72 | 4.47 | 5.19 | 4.65 | 5.41 | 4.71 | 4.19 |

**(c) Wind shear at 6–15 km a.s.l. (m/s/km) / Vertical resolution of RS is 300-m**

| | | | | | | | |
| --- | --- | --- | --- | --- | --- | --- | --- |
| RS | 8.30 | 9.50 | 9.41 | 7.72 | 9.80 | 9.38 | 8.00 |
| ERA5 | 4.00 | 5.22 | 5.84 | 5.21 | 6.14 | 4.76 | 3.37 |

**(d) Wind shear at 20–21 km a.s.l. (m/s/km) / Vertical resolution of RS is 400-m**

| | | | | | | | |
| --- | --- | --- | --- | --- | --- | --- | --- |
| RS | 9.02 | 10.40 | 11.67 | 12.56 | 12.14 | 10.48 | 9.80 |
| ERA5 | 3.00 | 3.83 | 4.79 | 5.59 | 4.72 | 3.63 | 2.98 |

2) paragraph 3.2: The authors correctly note that the frequency of occurrence (OF) of KHIs depends on the vertical resolution of the gradients. The fact that the OFs from the model coincide better with those from the HVRRS with a threshold Ri < 1 is therefore fortuitous since it depends on the resolution of the HVRRS. (If you had calculated the gradients on a 50 m scale, and not 10 m, you would have lower OFs, and therefore a different threshold to apply on the model estimates).

Can you please comment on this fact?

Response: We agree. Since the thickness of turbulence in the free troposphere is about 200-m based on our previous studies, the moving average number for a 50-m gridded profile would be 4. The $OF(Ri<1/4)$ for a 10-m resolution and 20 moving average

number is generally comparable with that for a 50-m resolution and 4 moving average number (Figure 1 in the updated file). The moving average in vertical is crucial to inhibit the instantaneous convection, as stated in Section 2.3.

In addition, as stated above, based on your comment we carried out another comparison experiment (ERA5 and radiosonde have a comparable vertical resolution) and also found that *Rit* should be larger than 1/4.

3) The authors systematically attribute Ri < 1/4 occurrences to KH instabilities. This is certainly not always the case. Thus, the diurnal boundary layer is very probably close to a neutral static stability (Ri ~ 0) without having anything to do with KH instabilities. The same is true in the tropical troposphere, where deep convective cells develops up to about 15 km altitude. The Ri < 1/4 occurrences are most likely a signature of an unstable flow, but not that this instability is due to a KHI. I recommend that the authors modify the discussion and conclusion paragraphs accordingly.
As well as the title of the article.
Response: We totally agree with you, very thanks for the suggestion! Based on your comments as well as the suggestion from Reviewer #1, we changed the title to "Occurrence frequency of subcritical Richardson number assessed by global high-resolution radiosonde and ERA5 reanalysis". Accordingly, in the updated files, the *OF*(KHI) has been replaced with *OF*(*Ri<Rit*) throughout all texts and figures.

4) Horizontal winds measured under radiosonde at the scale of a few tens of meters are affected by the chaotic movements of the gondola due to the pendulum and to the balloon's own movements (see for example Ingleby et al., 2022, https://doi.org/10.5194/amt-15-165-2022 and references therein). A low-pass filter is applied to the HVRRS profiles to reduce these effects. This filtering should have an impact on the effective resolution of the wind measurements (which is much larger than 10 m). Although it is difficult to assess the impact of this filtering, I suggest that you discuss this fact.
Response: Thanks! We have included this discussion in Section 2.3. In addition, we would like to address that the moving average in Eq.(1) could offset the effect of chaotic movements, to some extent, based on the finding in Figure 1.

**Specific comments**

- line 155: 115 million HVRRS profiles??? Do you confirm?
Response: Very sorry we make a serious mistake. We totally misunderstood the word "million" in the context of English for a long time (We thought it means 10,000). This analysis went through several versions. At the beginning of the analysis, about 1.15

million radiosonde was adopted during years 2016-2022. And then, we removed all data on 2016 due to the filled storage of our device (Since ERA5 137 model level data needs huge storage and computation resources). We rechecked our radiosonde dataset during years 2017-2022, and found that about 0.95 million radiosondes have been adopted.

The following Linux terminal displays the total count of radiosonde profiles during years 2017-2022.

```
jian@LAPTOP-2ECIDBLP:/mnt/d/RS_all_10m$ find . -name "*-201[7-9]*"  | wc -l
449531
jian@LAPTOP-2ECIDBLP:/mnt/d/RS_all_10m$ find . -name "*-202*"  | wc -l
495860
```

- Line 233: a "tropospheric segment" from 2 to 8.9 km is chosen. Why this choice (if interested in the OF(KHI) in the 0-2 and 10-15 km height ranges?

Response: The study height interval is from the near-ground up to 30 km. Our primary motivation was to select two typical height intervals (0-2 and 10-15 km) for visualization.

In Section 3.4, we have changed the subtitle to "The dynamical environment of $OF(Ri<Rit)$ in the free troposphere". The extracted gravity waves at 2-8.9 km were to characterize the wave environment of $OF(Ri<Rit)$ in the free troposphere.

- line 240 (Fig. 2): what is the vertical resolution of the shear estimates from HVRRS? Please, specify.

Response: In the updated files, we have stated that HVRRS-based wind shear is taken from Eq.(1), with a vertical resolution of 10-m.

- lines 259 & 282: Are the ERA5 shear estimates dramatically lower than the HVRRS estimates if the RS gradients are estimated with the same resolution as the model (i.e. 300 m for the 10-15 km altitude domain)?

Response: According to the information in Table 3 in the revised manuscript, under a comparable vertical resolution, ERA5-based wind shear at 6–15 km a.s.l. is underestimated by around 4 m/s/km. Moreover, the following statement has been inserted into the revised Section 3.2:

"…As illustrated in Table 3, even accounting for the fact that ERA5 has a comparable vertical resolution of radiosonde, wind shears in ERA5 reanalysis are still underestimated by around 51.9%, 50.7%, 44.5%, and 62.5% at 0.8–1.3 km, 2.2–3.2 km, 6–15 km and 20–21 km a.s.l., respectively…"

Moreover, In Conclusion section, one paragraph has been added:

"However, the vertical resolution of ERA5 reanalysis sharply decreases with altitude,

which is not comparable with HVRRS. Thus, to match with ERA5 reanalysis at specified height intervals, the HVRRS was vertically interpolated with resolutions spanning from 100-m to 400-m. Even at a comparable resolution, vertical wind shear is underestimated by around 50%, leading to a considerable underestimation in $OF(Ri<1/4)$, compared to radiosondes. "

- line 306: I agree with the statement that PBL is mixed by convection during daytime. Clearly, the occurrence of Ri < 1/4 is not attributable to KH instabilities in such cases.
Response: We agree. The mean $OF(0<Ri<Rit)$ in ERA5 reanalysis 0–2 km a.s.l. is significantly lower than that of $OF(Ri<Rit)$ (Figure S5a and Figure 9a). Around 40% of $OF(Ri<Rit)$ can be attributed to $OF(Ri<0)$ (Figure 8).

In addition, the related content has been modified correspondingly.

- line 376: this statement is not visible in figure 9.
Response: The statement has been removed, thanks.

- lines 381-3: I doubt that the Ri < 1/4 occurrences are only due to KHI in this region where deep convective cells are frequent.
Response: Yes. The percentage of $OF(Ri<0)$ relative to $OF(Ri<Rit)$ in the PBL is around 45%. We have added Figure 8 to illustrate the contribution from $Ri<0$, which is likely related to deep convective cell.

[Figure]

**Figure 8**. The percentage of $OF(Ri<0)$ relative to $OF(Ri<Rit)$ in HVRRS (red) and ERA5 reanalysis (blue).

- lines 414 & 793: la Niño → la Niña
Response: Amended.

- line 436: I do not understand your concluding sentence, figure 14b showing precisely that the probability of occurrence Ri < 1/4 depends almost not on the total energy of the gravity waves, but almost exclusively on the horizontal wind shear (if it exceeds 18 m/s/km).
Response: Figure 14 was intended to investigate the gravity wave characteristics of $OF(Ri<Rit)$. For a clearer presentation, we calculated the distribution of $OF(Ri<Rit)$ categorized by GW total energy, which is illustrated as Figure S9 in the revised supporting information. The $OF(Ri<Rit)$ obviously statistically increases with GW total energy.

[Figure]

**Figure S9**. Violin plot of $OF(Ri<Rit)$ categorized by GW total energy (a) and wind speed (b).

- line 443-456: I do not understand your conclusion about the wind speed. Figure 15 clearly shows that the occurrence Ri < 1/4 does not depend on the wind speed (if the wind speed exceeds a few m/s), but occurs with high probability if the shear exceeds ~20 m/s/km. This is a convincing result!
Response: Many thanks for the comment. The motivation of Section 3.4 was to

investigate the association of GWs and mean flows with $OF(Ri<Rit)$ by enhancing wind shears, as stated in the first paragraph of Section 3.4. As stated above, we also investigate the distribution of $OF(Ri<Rit)$ categorized by wind speed. The result in Figure S9b indicates that the $OF(Ri<Rit)$ also almost linearly increases with wind speed.

---

## Referee Report (RR1)

Review report on the paper "Occurrence frequency of subcritical Richardson number assessed by global high-resolution radiosonde and ERA5 reanalysis" by Shao et al., ACPD.

The authors responded to the questions raised by the reviewers, and improved the manuscript which is now more conclusive compared to the initial submission, both in the handling of the data as well as in the interpretation and description of the results. Still, I think a few clarifications are necessary, mainly of technical or linguistic character.

p.5 L. 121 What does "the last version of the ECMWF model" refer to? The latest reanalysis product? The IFS version?

p.11 L.289 According to Fig. S3 the static stability is not averaged from the surface to 30 km.

Figure 8: Out of curiosity, is the ERA5 distribution above 17 km altitude so irregular because the data density is so low for both $Ri<0$ and $Ri<Ri_t$? How meaningful is the top part of the plot then?

p.15 L.425ff I would expect the vertical resolution to be enhanced over mountainous areas, due to the surface-following hybrid sigma-pressure coordinates.

p.16 L.450 "Generally weak" compared to where? I am not sure that I would agree with this statement.

p.17 L.471 Just to be sure, do the unresolved orographic gravity waves (their dissipation) cause the low Richardson numbers, or do the unresolved orographic waves occur along with resolved orographic gravity waves which impact the occurrence of low Richardson numbers? Or both effects? Maybe rephrase the sentence to make it more clear how you interpret the results.

p.17 L.477 I find it a bit hard to follow the interpretation of Fig. 15.
$OF(Ri<Ri_t) > 10\%$ would be yellow and above in the colorscale, I see no direct connection to the wind speed threshold of 25 m/s.
The occurrence frequency $OF(Ri<Ri_t)$ depends mainly (directly) on wind shear, and the average wind shear (along with $OF(Ri<Ri_t)$) increases somewhat with the average wind speed. However, this is mainly evident in Fig. S9b and not in Fig. 15.
Maybe rephrase this paragraph.
p.19 L.523 would have to be adjusted accordingly.

Figure 12: I believe SDOR should have meter as unit.

---

## Referee Report (RR2)

Review report on the revised paper **"Occurrence frequency of subcritical Richardson number assessed by global high-resolution radiosonde and ERA5 reanalysis"** by Shao et al. submitted to the journal Amospheric Chemistry and Physics.

**Overview**

This new version of the article shows substantial improvements. The authors have responded in detail to questions and suggestions. The modification of the title is welcome, as are the analyses of RS with vertical resolutions close to those of ERA5. The climatological results appear interesting (vertical shears and occurrence frequency of Ri<0.25 in different climate zone).  On the other hand, I still have questions about the effective vertical resolution of the estimated gradients and related Richardson number (see below).  Also, I am circumspect about the comparison of RS/ERA5 occurrence frequencies, as the approximate agreement found with Ri[ERA5] ~ 1 depends on the resolution of the RS estimates. I think this two points needs to be clarified in the final version.

In view of the article's substantial improvements, I agree that the article should be published with the inclusion of the requested details.

**Major comments**

• Paragraph 2.3: The comparison of Ri estimates and shears from radiosondes with resolutions comparable to those of the model (Table 2), seems relevant. However, I still don't understand your estimate of the vertical gradient evaluated over 10 m and averaged over 200 m. Doesn't arithmetic averaging finite differences over 10 m in 200 m windows amount to estimating the finite difference over 200 m?

$$\frac{1}{n}\sum_{i=1}^{n}\frac{\Delta T_i}{\Delta z}=\frac{T_n-T_1}{n\Delta z}$$

If so, the gradients, and hence the Richardson numbers, are estimated from finite differences of 200 m! Such a conclusion is supported by your figure 1 as gradients estimated with a vertical resolution of 10 m averaged over 20 bins are very close to the gradients estimated with a resolution of 50 m averaged over 4 bins. Please, clarify this (important) point: are averaged finite differences representative of vertical gradients over 200 m or over 10 m?

• The fact that the frequency of occurrence of Ri<1 in ERA5 is climatologically consistent with that of Ri<1/4 in radiosondes is fortuitous, since it depends on the effective vertical resolution for Ri estimated from the RSs. For example, if we used a better resolution for RS gradients, say 30 m, we would have a higher frequency of occurrence of KHI, and therefore better agreement with a threshold of Ri < 1.5 or 2 from ERA5. I suggest you comment on this point.

**Specific comments**

– You state here and there (e.g. line 252) that the shear resolution is equal to 10m. Is it really the case  (because of the averaging procedure, see above). If so, the statement "For 10-m radiosondes, the moving average in a step of 200-m could offset the effect of chaotic movements, at least to some extent" (lines 200-202) is certainly inaccurate.

– I wonder about the relevance of taking into account the 0-2 km height interval in this climatological study. This altitude interval is representative of the diurnal atmospheric boundary layer (ABL) at low latitudes, but certainly not at high latitudes. Is it relevant to compare the same 0-2 km altitude interval in the Arctic and at the equator? (I don't think it is).

– Figure 11: please, use the same ranges for the x axis in order to help for a direct visual comparison.

---

## Author Response (AR2)

**Response to Reviewers #1 Comments**

**We thank the associate editor, editor and two anonymous reviewers again for their thoughtful and exhaustive comments and suggestions, which significantly help us to improve the quality of the manuscript. In this revised manuscript, we have revised the manuscript accordingly. Below, we indicate the original comment of the respective reviewer in blue and our point-to-point response is denoted in black.**

**Before addressing the comments, we would like to express our sincere gratitude to the reviewers for their exceptionally informative, constructive, and detailed comments.**

**Reviewer #1 Evaluations:**

Review report on the paper "Occurrence frequency of subcritical Richardson number assessed by global high-resolution radiosonde and ERA5 reanalysis" by Shao et al., ACPD.

The authors responded to the questions raised by the reviewers, and improved the manuscript which is now more conclusive compared to the initial submission, both in the handling of the data as well as in the interpretation and description of the results. Still, I think a few clarifications are necessary, mainly of technical or linguistic character.

Response: We sincerely thank the reviewer again for your professional and detailed comments. According to your suggestion, we have fixed errors caused by ERA5 coordinates throughout the texts, which will be discussed later. Another error we have fixed is Figure 9. The altitude in Fig.9b should be 5-10 km rather than 10-15 km (We forgot to inverse the model level of ERA5 in this program).

p.5 L. 121 What does "the last version of the ECMWF model" refer to? The latest reanalysis product? The IFS version?

Response: Sorry for the mistake. "last" should be replace by "latest". The sentence has

been rephrased as:

"…By comparison, ERA5 global reanalysis can provide a seamless coverage of temperature and wind, and it is the latest generation of the European Centre for Medium-Range Weather Forecasts (ECMWF) atmospheric reanalysis and is based on the state-of-the-art Integrated Forecasting System (IFS) Cy41r2 (Hersbach et al., 2020; Gu et al., 2023)…".

p.11 L.289 According to Fig. S3 the static stability is not averaged from the surface to 30 km.

Response: Sorry we forget to modified the main text in our last version. Now the sentence has been modified to be:

"…By comparison, the ERA5-acquired $N^2$ averaged over four height intervals (e.g., 0–5, 5–10, 10–15, 15–20 km a.g.l.) is reliably estimated at all heights…".

Figure 8: Out of curiosity, is the ERA5 distribution above 17 km altitude so irregular because the data density is so low for both Ri<0 and Ri<Rit? How meaningful is the top part of the plot then?

Response: The occurrence frequency of *Ri<Rit* for ERA5 is as low as 0.05% in the lower stratosphere (Tab. 4c), which can lead to the abrupt change in terms of *OF*(*Ri*<0)/ *OF*(*Ri*<*Rit*) in the lower stratosphere. The occurrence frequency of *Ri*<*Rit* above 17 km altitude could have potential implications for the investigation of clear air turbulence (CAT), which can be commonly observed in the upper troposphere and lower stratosphere (UTLS). In addition, ERA5 was also used for the study of upper-level turbulence encountered by cruising aircraft (for instance, Lee et al., 2023, JGR-Atmospheres). Also turbulence (or wind shear) in the UTLS have implications for constituent mixing across the tropopause (Lee et al., 2019, Nature). The present analysis can provide some information on the quantitative comparison between ERA5 and radiosonde in the UTLS region.

p.15 L.425ff I would expect the vertical resolution to be enhanced over mountainous

areas, due to the surface-following hybrid sigma-pressure coordinates.

Response: We agree. In the previous version, we transferred ERA5 model level to geopotential height based on the definition in https://confluence.ecmwf.int/display/UDOC/L137+model+level+definitions, which can lead to substantial errors in estimating terrestrial $OF(Ri<Rit)$ in the low troposphere. The IFS model level indeed follows hybrid sigma-pressure coordinates, and the calculation (model level to geopotential height) should follow the procedure posted on the ECMWF website (https://confluence.ecmwf.int/display/CKB/ERA5%3A+compute+pressure+and+geopotential+on+model+levels%2C+geopotential+height+and+geometric+height#ERA5:computepressureandgeopotentialonmodellevels,geopotentialheightandgeometricheight-Pressureonmodellevels). The geopotential is estimated based on the python program "compute_geopotential_on_ml.py".

The updated coordinates will lead to some changes in wind shears and $OF(Ri<Rit)$, mainly in the low troposphere. Therefore, we have recalculated all results throughout the text based on the hybrid sigma-pressure coordinate.

Thanks again for your very professional comments, which help us to fix a big technical error.

p.16 L.450 "Generally weak" compared to where? I am not sure that I would agree with this statement.

Response: The statement has been rephrased as:

"…In the free troposphere the percentage of $OF(Ri<0)$ relative to $OF(Ri<Rit)$ is generally less than 20% (Fig. 8), KHI is preferentially generated from strong wind shear…"

p.17 L.471 Just to be sure, do the unresolved orographic gravity waves (their dissipation) cause the low Richardson numbers, or do the unresolved orographic waves occur along with resolved orographic gravity waves which impact the occurrence of low Richardson numbers? Or both

effects? Maybe rephrase the sentence to make it more clear how you interpret the results.

Response: The present analysis can only imply the potential contribution from unresolved orographic waves. It is hard to quantify the effect of resolved orographic GWs on *Ri* here.

In Yasiui et al. (2018), resolved GWs in the MLT region was found to interact with wind shears. Also in Lachnitt et al. (2023), they stated that orographic waves lead to turbulent mixing in the troposphere and in the stratosphere.

However, it would be difficult for us to conclude the role of resolved orographic waves in present analysis. We feel sorry for that.

The above concern has been incorporated in the main text.

References:

Yasui, R., Sato, K., & Miyoshi, Y. (2018). The momentum budget in the stratosphere, mesosphere, and lower thermosphere. Part II: The in situ generation of gravity waves. Journal of the Atmospheric Sciences, 75(10), 3635-3651

Lachnitt, H.C., Hoor, P., Kunkel, D., Bramberger, M., Dörnbrack, A., Müller, S., Reutter, P., Giez, A., Kaluza, T., and Rapp, M.: Gravity-wave-induced cross isentropic mixing: a DEEPWAVE case study, Atmos. Chem. Phys., 23, 355–373, https://doi.org/10.5194/acp-23-355-2023, 2023.

p.17 L.477 I find it a bit hard to follow the interpretation of Fig. 15. OF(Ri<Rit) > 10% would be yellow and above in the colorscale, I see no direct connection to the wind speed threshold of 25 m/s. The occurrence frequency OF(Ri<Rit) depends mainly (directly) on wind shear, and the average wind shear (along with OF(Ri<Rit)) increases somewhat with the average wind speed. However, this is mainly evident in Fig. S9b and not in Fig. 15. Maybe rephrase this paragraph. p.19 L.523 would have to be adjusted accordingly.

Response: We agree. *OF(Ri<Rit)>*10% can be frequently observed when wind shear is larger than 20 m/s/km, rather than wind speed exceeding 25 m/s. The correction has been made in the main text.

Figure 12: I believe SDOR should have meter as unit.

Response: According to the ERA5 website posted on https://cds.climate.copernicus.eu/cdsapp#!/dataset/reanalysis-era5-single-levels?tab=overview, SDOR is dimensionless.

**Response to Reviewers #2 Comments**

**We thank the associate editor, editor and two anonymous reviewers again for their thoughtful and exhaustive comments and suggestions, which significantly help us to improve the quality of the manuscript. In this revised manuscript, we have revised the manuscript accordingly. Below, we indicate the original comment of the respective reviewer in blue and our point-to-point response is denoted in black.**

**Before addressing the comments, we would like to express our sincere gratitude to the reviewers for their exceptionally informative, constructive, and detailed comments.**

**Reviewer #2 Evaluations:**

Review report on the revised paper "Occurrence frequency of subcritical Richardson number assessed by global highresolution radiosonde and ERA5 reanalysis" by Shao et al. submitted to the journal Amospheric Chemistry and Physics.

Overview This new version of the article shows substantial improvements. The authors have responded in detail to questions and suggestions. The modification of the title is welcome, as are the analyses of RS with vertical resolutions close to those of ERA5. The climatological results appear interesting (vertical shears and occurrence frequency of Ri

Response: We sincerely appreciate the reviewer again for your patient and insightful comments and assessments. According to your suggestion, we have addressed the issue concerning radiosonde resolution in the revised version.

Major comments

• Paragraph 2.3: The comparison of Ri estimates and shears from radiosondes with resolutions comparable to those of the model (Table 2), seems relevant. However, I still

don't understand your estimate of the vertical gradient evaluated over 10 m and averaged over 200 m. Doesn't arithmetic averaging finite differences over 10 m in 200 m windows amount to estimating the finite difference over 200 m?

$\frac{1}{n}\sum_{i=1}^{n}\frac{\Delta T_i}{\Delta z} = \frac{T_n - T_1}{n\Delta z}$

If so, the gradients, and hence the Richardson numbers, are estimated from finite differences of 200 m! Such a conclusion is supported by your figure 1 as gradients estimated with a vertical resolution of 10 m averaged over 20 bins are very close to the gradients estimated with a resolution of 50 m averaged over 4 bins. Please, clarify this (important) point: are averaged finite differences representative of vertical gradients over 200 m or over 10 m?

Response: The calculation of $Ri$ was handled over a vertical gradient of 10-m. While a moving average was previously applied to wind shear and buoyancy frequency. The averaged parameter at altitude $i$ can be represented as:

$$\overline{A}(i) = \frac{1}{n}\sum_{j=i-10}^{i+10} A(j)$$

where A demotes wind shear or Brunt-Väisälä frequency and n is the number of vertical bin.

Since 10-m radiosonde variables can be highly polluted by measurement noises, a moving average would be a necessary. In the following text, we will address this issue in more detail.

The above statement has been incorporated in the main text.

• The fact that the frequency of occurrence of Ri<1 in ERA5 is climatologically consistent with that of Ri<1/4 in radiosondes is fortuitous, since it depends on the effective vertical resolution for Ri estimated from the RSs. For example, if we used a better resolution for RS gradients, say 30 m, we would have a higher frequency of occurrence of KHI, and therefore better agreement with a threshold of Ri < 1.5 or 2 from ERA5. I suggest you comment on this point.

Response: The variation of buoyancy frequency and wind shear is strongly influenced by turbulence fluctuations and measurement noises. For instance, in Fig. 3d of Kantha

& Hocking (2011), turbulence can be frequently observed at almost all heights (Thorpe scale greater 0 can be roughly taken as the occurrence of turbulence). Without a moving average, many of the square of the buoyancy frequencies will be less than 0 for a 10-m resolution radiosonde, especially in the boundary layer.

The outer scale of turbulence is about few hundred meters in the boundary layer (Solanki et al., 2022), and then decreases to around 100 m in the troposphere (Rao et al., 2001). For large-eddy simulations, the spatial resolution for turbulence study typically ranges from around 5-m to 100-m, for instance, Schulte et al. (2022), Schalkwijk et al. (2015), Verrelle et al. (2017), and Strauss et al. (2022). In addition, for 10-m radiosonde, measurements noises are a big problem (more information refers to Wilson & Luce, 2011). In Wilson & Luce (2011), they split the profile in segment of 200 m to estimate noises. Therefore, we applied a 200-m moving average procedure to inhabit the effect from turbulence fluctuations and measurement noises.

Without a smoothing in vertical, a higher resolution generally leads to a larger occurrence frequency of $Ri<Rit$. For example, the averaged occurrence frequency of $Ri<Rit$ at 10-15 km a.g.l. is 5.29% for the ERA5 reanalysis, while it is as high as 30% for 10-m radiosondes. In this case, the threshold $Ri$ for the ERA5 reanalysis will even exceed 3, to produce a comparable $OF(Ri<Rit)$ with 10-m radiosondes.

Moreover, in the conclusion section, we have added a phrase to address the limitation of present analysis:

"... It is worth highlighting that HVRRS experiences a 200-m vertical moving average procedure to inhabit measurement noises and turbulence fluctuations. Without this procedure, the threshold $Ri$ for the ERA5 reanalysis would even higher than 1."

References:

Solanki, R., Guo, J., Lv, Y., Zhang, J., Wu, J., Tong, B., & Li, J. (2022). Elucidating the atmospheric boundary layer turbulence by combining UHF radar wind profiler and radiosonde measurements over urban area of Beijing. Urban Climate, 43, 101151.

Rao, D. N., Rao, T. N., Venkataratnam, M., Thulasiraman, S., Rao, S. V. B., Srinivasulu,

P., & Rao, P. B. (2001). Diurnal and seasonal variability of turbulence parameters observed with Indian mesosphere-stratosphere-troposphere radar. Radio Science, 36(6), 1439-1457.

Schulte, R. B., van Zanten, M. C., van Stratum, B. J. H., and Vilà-Guerau de Arellano, J.: Assessing the representativity of NH3 measurements influenced by boundary-layer dynamics and the turbulent dispersion of a nearby emission source, Atmos. Chem. Phys., 22, 8241–8257, https://doi.org/10.5194/acp-22-8241-2022, 2022.

Schalkwijk, J., H. J. J. Jonker, A. P. Siebesma, and F. C. Bosveld, 2015: A year-long large-eddy simulation of the weather over Cabauw: An overview. Mon. Wea. Rev., 143, 828–844, doi:10.1175/MWR-D-14-00293.1.

Verrelle, A., D. Ricard, and C. Lac, 2017: Evaluation and improvement of turbulence parameterization inside deep convective clouds at kilometer-scale resolution. Mon. Wea. Rev., 145, 3947–3967, https://doi.org/10.1175/MWR-D-16-0404.1.

Strauss, C., Ricard, D., & Lac, C. (2022). Dynamics of the cloud–environment interface and turbulence effects in an LES of a growing cumulus congestus. Journal of the Atmospheric Sciences, 79(3), 593-619.

Kantha, L., & Hocking, W. (2011). Dissipation rates of turbulence kinetic energy in the free atmosphere: MST radar and radiosondes. Journal of Atmospheric and Solar-Terrestrial Physics, 73(9), 1043-1051.

Wilson, R., Dalaudier, F., and Luce, H.: Can one detect small-scale turbulence from standard meteorological radiosondes?, Atmos. Meas. Tech., 4, 795–804, https://doi.org/10.5194/amt-4-795-2011, 2011.

[Figure]

**Fig. 3.** (a) Cumulative displacement (m); (b) Trend to noise ratio (TNR); (c) Thorpe displacement (m); (d) Thorpe scale (m); and (e) TKE dissipation rate (W/kg) for the sonde 070623B (#8) released on June 23, 2007. In panel (b), the blue line shows TNR=1.5, and the red lines show the bulk TNR in the troposphere and the stratosphere. (For interpretation of the references to colour in this figure legend, the reader is referred to the web version of this article.)

Figure A. Figure 3 in Kantha & Hocking (2011).

**Specific comments**

– You state here and there (e.g. line 252) that the shear resolution is equal to 10m. Is it really the case (because of the averaging procedure, see above). If so, the statement "For 10-m radiosondes, the moving average in a step of 200-m could offset the effect of chaotic movements, at least to

some extent" (lines 200-202) is certainly inaccurate.

Response: Thanks for the suggestion. We have removed this statement in the revised draft. It would be difficult to assess the chaotic movement in this study. The chaotic movement may be characterized by the accuracy of wind speed. Different types of radiosonde can have various accuracies. For instance, Vaisala-92 has an accuracy of around ±0.2 m/s (Wang et al., 2020), Vaisala PTB201A has an accuracy of ±2% and 5° for wind speed and wind direction, respectively (Conroy et al., 2016), and WXT510 Vaisala has an accuracy of ±0.3 m/s (Tratt et al., 2011). However, it is also difficult for us to address the accuracy of all soundings due to the near-global distribution of radiosonde. In addition, we have included the following statement to

address this issue:

"…However, it is hard to quantify the movement in present study…"

References:

Wang, D., Guo, J., Chen, A., Bian, L., Ding, M., Liu, L., et al. (2020). Temperature inversion and clouds over the Arctic Ocean observed by the 5[th] Chinese National Arctic Research Expedition. Journal of Geophysical Research: Atmospheres, 125, e2019JD032136. https://doi.org/10.1029/2019JD032136

Conroy, J. L., D. Noone, K. M. Cobb, J. W. Moerman, and B. L. Konecky (2016), Paired stable isotopologues in precipitation and vapor: A case study of the amount effect within western tropical Pacific storms, J. Geophys. Res. Atmos., 121, 3290–3303, doi:10.1002/2015JD023844.

Tratt, D. M., S. J. Young, D. K. Lynch, K. N. Buckland, P. D. Johnson, J. L. Hall, K. R. Westberg, M. L. Polak, B. P.Kasper, and J. Qian (2011), Remotely sensed a mmonia emission from fumarolic vents associated with a hydrothermally activ efault in the Salton Sea Geothermal Field, California, J. Geophys. Res., 116, D 21308, doi:10.1029/2011JD016282.

– I wonder about the relevance of taking into account the 0-2 km height interval in this climatological study. This altitude interval is representative of the diurnal atmospheric boundary layer (ABL) at low latitudes, but certainly not at high latitudes. Is it relevant to compare the same 0-2 km altitude interval in the Arctic and at the equator? (I don't think it is).

Response: PHL depth strongly varies with local time, latitude, season, land cover, etc. Also, algorithms can arise large uncertainties in estimating PHL depth. Its variation is a complex topic. PHL depth in the tropical can be quite different with that of polar regions. Therefore, we have referred the 0-2 km altitude as the low troposphere throughout the text, instead of PBL.

– Figure 11: please, use the same ranges for the x axis in order to help for a direct

visual comparison.

Response: Amended as suggested, thanks.

---

## Author Response (AR3)

**Response to the Editor/Technical Comments**

Dear Authors,

I am overall convinced by the changes made to the manuscript in response to the reviews. However, related to the comment by referee 2, the explanation of the averaging (sect 2.3) could be further clarified. I see two possible ways to understand the current description:

- the shear and Brunt Vaisala frequency are computed at 10 m resolution (which implies a square root), and then those estimates are averaged over 200 m (20 points) and squared. I think this is what is done in light of the following paragraphs, in which case please state it unambiguously after Eq. 1.

or – the squared shear and BVF are computed and averaged over 200m, which would be strictly equivalent to subsampling at 200 m.

After this point has been clarified, I expect that your paper will be suitable for publication in ACP.

Sincerely,
Aurelien Podglajen

Response: Very thanks for the comments. Per your suggestion, we have stated that wind shear and Brunt-Väisälä frequency are computed at 10 m resolution in Equation 1. The related sentences after Eq.1 have been rephrased as:

"…Therefore, the wind shear and Brunt-Väisälä frequency are computed at 10 m resolution, and then those estimates are averaged over 200 m (20 points) and squared. More exactly, the averaged parameter at altitude $i$ can be represented as $\overline{A}(i) = \frac{1}{n}\sum_{j=i-10}^{i+10} A(j)$, where $A$ denotes wind shear or Brunt-Väisälä frequency and $n$ is the number of vertical bin…"

**Notification to the authors**:

1. It seems that tables are included as figures #15, S6, S8. If it is so, they must be re-labelled as tables and the references in the manuscript text must be adjusted accordingly. A table may be inserted as an image, but still be called as a table. 2. Please ensure that the colour schemes used in your maps and charts allow readers with colour vision deficiencies to correctly interpret your findings. Please check your figures using the Coblis – Color Blindness Simulator (https://www.color-blindness.com/coblis-color-blindness-simulator/) and revise the colour schemes accordingly.

Response: Thanks for the validation. Figures 15, S6, S8 are heat maps and inserted as figures, rather than tables. Moreover, all color bars have been set to rainbow color schemes, and Figure 10 has been replaced with complete color bar.